# STORM: Segment, Track, and Object Re-Localization from a Single Image

**Yu Deng** [* 1 2]  **Teng Cao** [* 1 2]  **Hikaru Shindo** [1]  **Quentin Delfosse** [† 1 3]  **Jiahong Xue** [1]  **Kristian Kersting** [1 2 4 5]

## Abstract

Accurate 6D pose estimation and tracking are core capabilities for physical AI systems, yet real-world deployment remains brittle and labor-intensive. Many pipelines rely on CAD models, manual masking, or per-object adaptation, and still fail under occlusion or fast motion without a principled way to recognize failure. We propose STORM, a unified framework for reference-conditioned 6D tracking that can operate from a single reference image, with minimal manual input and improved robustness. STORM combines: (i) Hierarchical Spatial Fusion Attention (HSFA), a task-driven reference-query fusion architecture that supports both single-reference and multi-reference conditioning and can optionally use vision-language semantic conditioning to resolve instance ambiguities; and (ii) a BCE-trained tracking verifier whose continuous compatibility logit is used as an energy-like score to detect drift and trigger automatic re-initialization. Experiments on LM-O and YCB-Video show that STORM improves annotation-free pose tracking accuracy over strong baselines and recovers reliably from severe occlusions and rapid viewpoint changes with minimal overhead.

## 1. Introduction

Zero-shot object pose estimation and tracking in dynamic, unstructured environments represent a fundamental chal-

---

[*]Equal contribution [1]Department of Computer Science, Technical University of Darmstadt, Darmstadt, Hesse, Germany [2]Hessian Center for Artificial Intelligence (hessian.AI), Darmstadt, Hesse, Germany [3]Google Intrinsic AI Research, Germany. † Work done while at the AIML research lab, now working at Intrinsic, Google. [4]German Research Center for Artificial Intelligence (DFKI), Darmstadt, Hesse, Germany [5]Centre for Cognitive Science, Technical University of Darmstadt, Darmstadt, Hesse, Germany. Correspondence to: Yu Deng <yu.deng@tu-darmstadt.de>, Teng Cao <teng.cao@tu-darmstadt.de>, Kristian Kersting <kersting@tu-darmstadt.de>.

*Proceedings of the 43rd International Conference on Machine Learning*, Seoul, South Korea. PMLR 306, 2026. Copyright 2026 by the author(s).

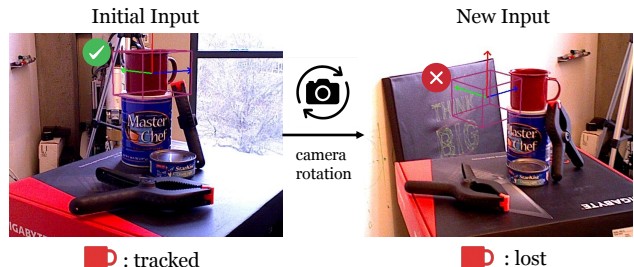

*Figure 1.* **Pose-estimation models often lack robustness**, here exemplified with FoundationPose (Wen et al., 2024), that fails to detect a mug under camera pose variation, highlighting its sensitivity to viewpoint shifts. Best viewed in color.

lenge in machine learning and computer vision. The core difficulty lies in the severe distributional shift between the canonical reference information (e.g., a single clean image or a 3D model) and the observational queries (Hodaň et al., 2020; Nguyen et al., 2023), which are often plagued by heavy occlusion, motion blur, and drastic viewpoint changes. While accurate 6D pose estimation is a prerequisite for embodied agents (Kappler et al., 2018; Wen et al., 2022), achieving robustness without extensive instance-specific training or manual annotation remains an open problem.

Despite recent advances (He et al., 2020; Wen et al., 2024), current methods face critical limitations rooted in their learning paradigms. First, they typically rely on explicit 3D priors (e.g., curated CAD models) to bypass the representation learning gap, limiting direct deployment when inference-time object models are unavailable. Second, reference-based approaches like CNOS (Nguyen et al., 2023) and PerSAM (Zhang et al., 2023) often employ standard visual encoders with simple metric learning objectives (e.g., cosine similarity). These metrics are inherently brittle; they fail to capture the non-linear manifold deformations induced by cluttered backgrounds or rapid rotations (Hodaň et al., 2020), leading to tracking divergence. Crucially, existing trackers often operate "blindly": they lack a built-in tracking-validity signal to detect when the tracking hypothesis has significantly drifted from the target object, making autonomous recovery difficult (see Figure 1).

To address these challenges, we propose **STORM** (Segment, Track, and Object Re-localization from a Single iMage), a unified framework that bridges semantic understanding and

geometric consistency. Unlike traditional pipelines that treat segmentation and tracking as separate engineering modules, STORM integrates them through two coupled modules: (1) Hierarchical Reference-Query Fusion: To resolve the domain gap between reference and query views, we introduce Hierarchical Spatial Fusion Attention (HSFA). Instead of simple feature concatenation, HSFA repeatedly aggregates reference-view evidence and query-reference interactions, with optional semantic conditioning from Vision-Language Models (VLMs) (Radford et al., 2021) to disambiguate visually similar instances. (2) Tracking-Loss Verification: TOM is trained as a binary compatibility classifier over current observations and a memory pool of successful tracks. At inference time, we convert its logit into an energy-like continuous score, smooth it temporally, and trigger re-localization only after persistent evidence of drift.

We empirically demonstrate that STORM achieves strong performance on challenging benchmarks, showing that reference-conditioned representation learning together with explicit tracking-loss verification outperforms traditional template matching in annotation-free inference settings.

Overall, our key contributions are:

- We propose STORM[1], a **unified framework for annotation-free, reference-conditioned 6D pose tracking** that addresses the distributional shift between reference and query data. By bridging semantic understanding with geometric consistency, STORM does not require curated CAD models or instance-specific fine-tuning at inference time.
- We introduce **Hierarchical Spatial Fusion Attention** (HSFA), a task-driven architecture for variable-view reference-conditioned segmentation. HSFA combines reference-view aggregation, query-reference cross-attention, and optional VLM-derived semantic conditioning to improve robustness under clutter and occlusion.
- We formulate tracking verification as a **BCE-trained compatibility verification problem**. Implemented via the Tracking Object Module (TOM), this mechanism uses an energy-like score derived from the compatibility logit to detect persistent drift and trigger automatic re-initialization.
- We establish a **Tracking Failure Benchmark** that evaluates trackers not just on pose accuracy, but also on their ability to distinguish valid tracks from failure modes. Our experiments show that TOM provides more reliable tracking-loss detection than fixed metric-learning baselines.

**Conflict of Interest Disclosure.** This work was supported by public research grants from the European Union, the Federal Ministry of Research, Technology and Space within the JUPITER AI Factory, and the Federal Ministry of Food and Agriculture within the FarmerSpaceAI project, as detailed in the Acknowledgements. The authors declare no financial or other substantive conflicts of interest related to this work.

Let us now discuss the literature related to STORM.

## 2. Background and Related Work

STORM operates at the intersection of vision-language segmentation, object-centric 3D priors, and uncertainty-aware tracking. To contextualize our contributions, we examine how prior works approach the challenges of feature alignment and tracking reliability.

### 2.1. Reference-Conditioned Segmentation

Reference-based segmentation requires aligning a canonical reference to a cluttered, occluded query view under strong appearance and viewpoint changes. Early supervised pipelines (e.g., Mask R-CNN (He et al., 2017)) learn such alignment implicitly, but do not generalize in a zero-shot manner. Recent foundation models such as SAM3 (Carion et al., 2025) and DINOv3 (Siméoni et al., 2025) provide rich semantics and strong region proposals, yet they are primarily designed for generic promptable perception (points/text) and lack an explicit mechanism for *reference-conditioned* segmentation, *i.e.* leveraging a specific reference image or geometric prior to resolve instance ambiguity in clutter.

State-of-the-art zero-shot methods, including CNOS (Nguyen et al., 2023) and PerSAM (Zhang et al., 2023), partially bridge this gap by using pre-trained encoders, but they often rely on shallow metric learning (e.g., cosine similarity), which can be brittle under non-linear feature distortions induced by heavy occlusion and illumination changes. Methods that more tightly couple segmentation with 3D pose, such as SAM-6D (Lin et al., 2024) and Pos3R (Deng et al., 2025), still face practical limitations: frame-wise processing without temporal consistency can lose track under rapid motion, and explicit 3D-to-2D keypoint matching becomes fragile when geometric cues are sparse or occluded.

In contrast, STORM introduces Hierarchical Spatial Fusion Attention (HSFA) as a task-specific reference-query fusion architecture. HSFA adapts established attention operations to the setting where the number of reference views varies at inference time: reference tokens are first aggregated into an object-centric representation, query tokens then attend to this representation, and optional VLM-derived semantic priors modulate the visual features when language cues are available. This design gives SOM a learned alternative to fixed cosine matching while keeping HSFA tailored to reference-conditioned segmentation.

---

[1]Code available at https://github.com/YuDeng321/STORM.

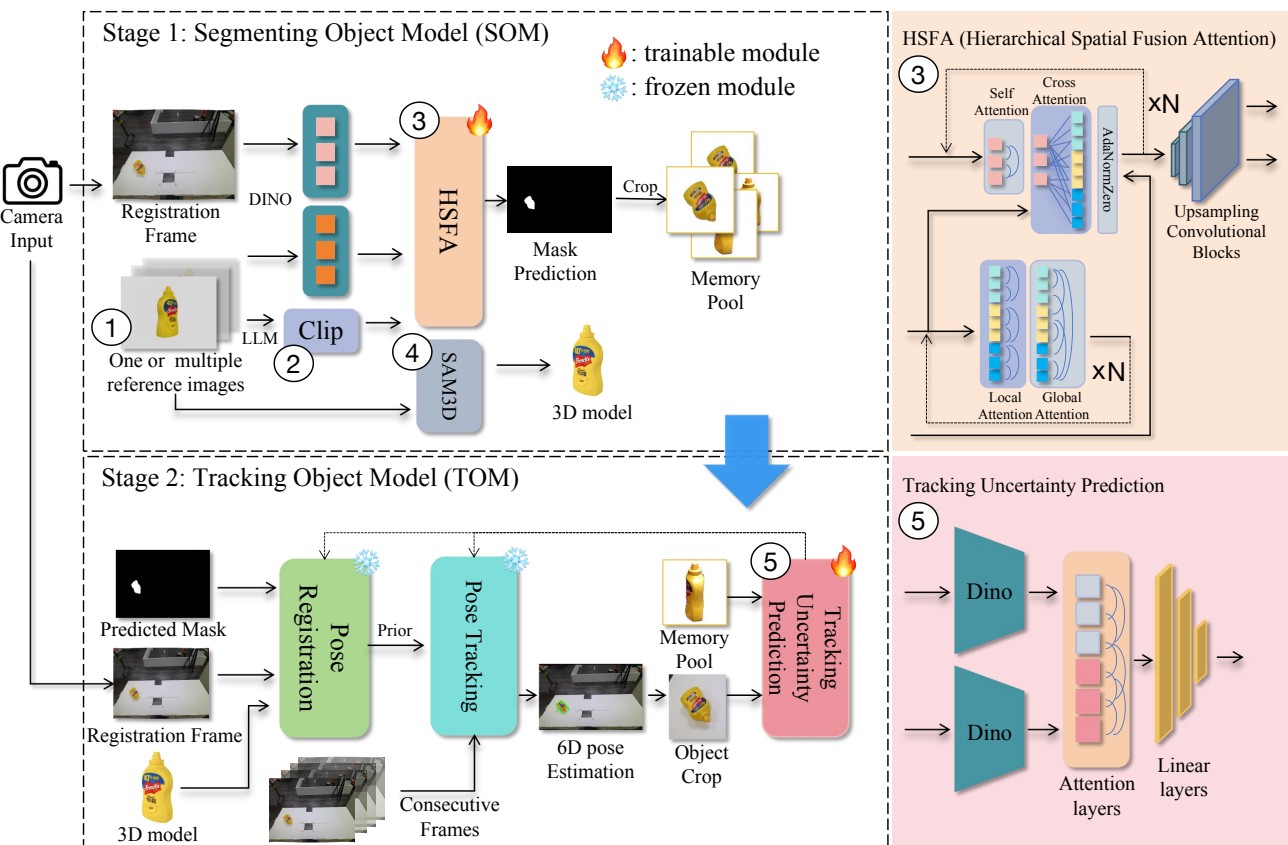

*Figure 2.* **Overview of the STORM Framework.** STORM unifies semantic understanding and geometric consistency via two learning-based modules. (Top) Segmenting Object Module (SOM): Instead of simple template matching, SOM employs Hierarchical Spatial Fusion Attention (HSFA) to aggregate variable reference views and align them with dense query features. VLM-derived semantic priors (via CLIP/LLM) are optional and help resolve spatial ambiguities when visual evidence is insufficient. (Bottom) Tracking Object Module (TOM): TOM is trained as a binary compatibility verifier between the current observation and a memory pool of successful tracks. Its logit is converted into an energy-like score; persistently high scores indicate tracking drift and trigger re-initialization. Best viewed in color.

## 2.2. Geometric Priors for Pose Estimation

Although 2D foundation models provide rich semantics, accurate pose estimation ultimately requires geometric consistency. Classical multi-view reconstruction (Cremers & Kolev, 2010) and NeRF-style approaches (Mildenhall et al., 2021; Muller et al., 2022) typically depend on dense multi-view supervision or costly per-scene optimization, which is often impractical for real-time, zero-shot deployment.

Single-image mesh prediction methods (e.g., Direct3D-S2 (Wu et al., 2025)) can recover geometry efficiently from one view, but the resulting meshes may be noisy and lack stable appearance-aligned cues, making downstream registration brittle when geometry is sparse or partially occluded.

We leverage SAM3D (Team et al., 2025b) to generate a canonical mesh from reference images and use it as a geometric anchor. Rather than relying on hard texture/geometry matching, we treat the mesh as a structural coordinate frame that supports lifting 2D semantics into 3D and enforce geometric consistency as a *soft* latent constraint, improving robustness when visual evidence is ambiguous or reconstruction quality is imperfect.

## 2.3. Tracking-Loss Verification

SOTA 6D pose trackers such as FoundationPose (Wen et al., 2024) provide strong local refinement but are typically "blind" to failures, implicitly assuming the target remains within a local basin; as a result, rapid motion or occlusion can cause silent drift, and ad-hoc recovery heuristics (e.g., particle filters (Deng et al., 2021) or histogram matching (Tjaden et al., 2017)) are often prone to false positives.

We instead learn an explicit verifier for tracking loss. TOM is trained with binary cross-entropy to distinguish compatible observation-memory pairs from identity-confusion and drift-like pairs. Motivated by continuous-score thresholding used in OOD detection (Liu et al., 2020), we use the negative compatibility logit as an energy-like score for temporal smoothing, thresholding, and re-initialization.

# 3. STORM

We propose STORM, a unified framework for annotation-free, reference-conditioned 6D pose tracking that addresses the fundamental challenge of distributional shift between canonical reference data and dynamic observational queries. Instead of relying on brittle template matching or explicit CAD priors at inference time, STORM formulates the task as a joint problem of reference-query feature fusion and tracking-loss verification. As illustrated in Figure 2, the framework operates in two coupled stages:

(1) The **Segmenting Object Module (SOM)** performs coarse-to-fine feature alignment to recover object masks and 3D geometry from reference image(s);

(2) The **Tracking Object Module (TOM)** functions as a learned compatibility verifier that monitors tracking fidelity and triggers re-initialization when drift persists.

STORM also makes the boundary between inherited and trainable components explicit. We use frozen foundation components for representation and geometry: DINOv3 provides dense visual features, the VLM/CLIP stack provides optional object-level text conditioning, SAM3D generates the reference mesh, and FoundationPose performs downstream pose registration/tracking. The trainable STORM components are SOM, including HSFA and the segmentation heads, and TOM, including the lightweight compatibility verifier used for failure detection.

## 3.1. SOM: Hierarchical Reference-Query Fusion

The core objective of SOM is to learn a mapping function $\mathcal{F}$ that aligns a query image $I_q$ (subject to occlusion and SE(3) transformation) with one or more reference views $I_{ref}$. We achieve this with Hierarchical Spatial Fusion Attention (HSFA), a task-driven composition of single-/multi-view reference aggregation, query-reference cross-attention, and optional semantic conditioning.

### 3.1.1. SEMANTIC PRIOR INJECTION.

To resolve spatial ambiguities in the latent space (e.g., distinguishing a target mug from a distractor), we inject semantic constraints derived from Vision-Language Models. We feed the reference $I_{ref}$ and a generic prompt $p$ into a VLM (Team et al., 2025a) to generate a concise object descriptor $T$. This text is encoded by CLIP into a semantic embedding $e_t \in \mathbb{R}^d$. Crucially, rather than simple concatenation, we treat $e_t$ as a conditioning variable that modulates the normalized visual tokens. The release model uses a zero-initialized AdaLN/FiLM-style conditioning layer to inject $e_t$ into the feature tokens $F \in \mathbb{R}^{N \times C}$:

$$\hat{F}_{i,c} = \left(1 + s_c(e_t)\right) \left(\frac{F_{i,c} - \mu_i}{\sigma_i + \epsilon}\right) + b_c(e_t), \quad (1)$$

where $\mu_i$ and $\sigma_i$ are computed over channels for token $i$, and $s_c(e_t)$ and $b_c(e_t)$ are zero-initialized scale and shift projections learned from $e_t$. Thus the language path is initially identity-preserving and learns residual feature-statistic corrections during training. In HSFA cross-attention, a separate sigmoid gate modulates reference key/value channels from the same language context, allowing the model to attenuate irrelevant reference channels when text is informative.

### 3.1.2. HIERARCHICAL SPATIAL FUSION ATTENTION (HSFA).

To bridge the domain gap between the clean reference and the cluttered query, HSFA performs alignment across multiple scales. Unlike standard cross-attention, HSFA fuses intra-view and inter-view dependencies (*cf.* Figure 2 ③):

- **Latent Reference Construction:** We process single- or multi-view reference patches via self-attention to build a coherent canonical representation $\mathcal{Z}_{ref}$.

- **Query-Reference Alignment:** The query features $\mathcal{Z}_{query}$ attend to $\mathcal{Z}_{ref}$ via cross-attention. In early layers, attention is computed against raw reference features to anchor global semantics; in deeper layers, it attends to refined spatial features to resolve local geometric details.

- **Iterative Refinement:** This fusion block is repeated $n$ times, progressively refining the query representation with reference-conditioned evidence.

### 3.1.3. IMPLICIT ALIGNMENT VIA HSFA.

HSFA induces a dense token-to-token correspondence between the query and the reference via an attention mechanism. Concretely, the cross-attention layer computes attention weights (row-wise normalized similarities, implemented with a softmax), which we interpret as an alignment matrix $W$. We then propagate a reference objectness prior through $W$ to obtain the query mask, and supervise the final mask with standard segmentation losses. Importantly, we do not impose an explicit correspondence loss; alignment emerges implicitly through mask supervision.

### 3.1.4. GEOMETRIC ANCHORING VIA SAM3D.

To lift the aligned 2D features into SE(3), we utilize SAM3D to generate a canonical 3D mesh $\mathcal{P}_{ref}$ from the reference image(s) (Figure 2 ④). This mesh serves as a rigid geometric anchor, enabling the downstream pose estimator to map the SOM-predicted masks onto metric 3D coordinates.

## 3.2. TOM: Tracking-Loss Verification

Standard pose trackers (e.g., FoundationPose (Wen et al., 2024)) rely on temporal continuity, which breaks under

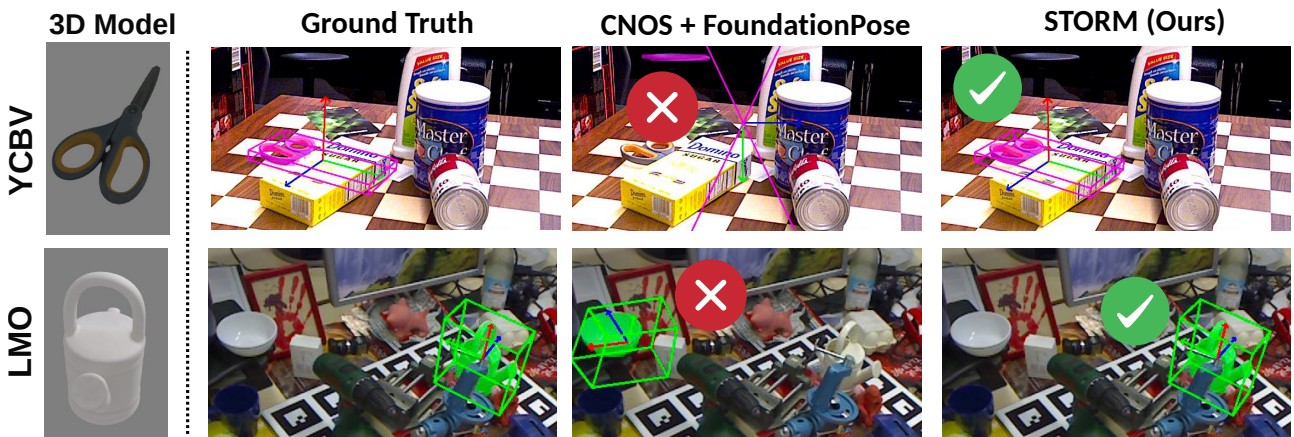

*Figure 3.* **STORM (SOM+TOM) achieves robust pose estimation for occluded objects in complex scenes.** We compare pose-estimation qualities on the LMO and YCB-V datasets, which comprise complex scenes with multiple and possibly occluded objects. As baselines, CNOS and GroundTruth are used to predict the segmentation mask, and FoundationPose was used to produce the pose estimation. The results indicate that our method produces pose estimates that are quantitatively and qualitatively close to the ground truth annotations in these scenarios. Detected objects are highlighted in green and pink. Best viewed in color.

rapid motion or heavy occlusion. Crucially, these methods lack a built-in mechanism to estimate tracking validity: they cannot distinguish between a difficult but recoverable frame and a complete tracking failure. We therefore train TOM as a binary compatibility verifier. Given a current observation crop $x_t$ and a memory pool $\mathcal{M}$ of previously successful crops, TOM predicts whether the current observation remains compatible with the tracked object. At inference time, we use the continuous verifier logit as a failure score rather than only using a hard binary decision.

### 3.2.1. COMPATIBILITY SCORE AND ENERGY-LIKE DECISION RULE.

We maintain a dynamic memory pool $\mathcal{M}$ containing real crops from successful past frames. For a new frame $t$, we extract visual features $\phi(x_t)$ using the DINOv3 backbone and match them against memory features. Instead of a fixed similarity metric, we employ a lightweight attention network $g_\theta(\cdot)$ that outputs a compatibility logit. Motivated by continuous energy-style scoring (LeCun et al., 2006; Liu et al., 2020), we define an energy-like score as the negative compatibility:

$$E(x_t, \mathcal{M}) \triangleq -g_\theta(x_t, \mathcal{M}). \tag{2}$$

Intuitively, a high compatibility logit generally implies a stable match, thereby resulting in low energy; conversely, drift yields low compatibility and high energy.

**From Energy to a Loss Decision.** Since $E = -g_\theta$, thresholding on energy is equivalent to thresholding on the logit: $E(x_t, \mathcal{M}) > \tau \Leftrightarrow g_\theta(x_t, \mathcal{M}) < -\tau$ (and equivalently on probability $\sigma(g_\theta) < \sigma(-\tau)$). We use an EMA-smoothed energy $\tilde{E}_t$ and declare tracking loss if $\tilde{E}_{t-k} > \tau$

for all $k = 0, \ldots, L-1$; in all experiments, $L = 3$. The threshold $\tau$ is calibrated once on a held-out verification split as the 95th percentile of compatible-pair scores, and the memory pool is a FIFO queue with $K = 16$ crops that is reset after re-localization and updated only from high-confidence tracked frames. Full state transitions are provided in Appendix H.1.

### 3.2.2. BCE TRAINING OBJECTIVE.

We train TOM on a synthetically generated verification dataset. We treat ground-truth-compatible observation-memory pairs as positive data $\mathcal{D}_{in}$, and generate incompatible pairs $\mathcal{D}_{out}$ using identity confusion and drift-like random crops. The optimization objective is binary cross-entropy over the compatibility logit:

$$\mathcal{L}_{\text{TOM}} = -\mathbb{E}_{(x, \mathcal{M}, y)} \big[ y \log \sigma(g_\theta(x, \mathcal{M})) \\ + (1-y) \log(1 - \sigma(g_\theta(x, \mathcal{M}))) \big], \tag{3}$$

where $y = 1$ denotes a compatible observation-memory pair and $y = 0$ denotes a drift or identity-confusion pair. This training increases the margin between compatible and incompatible pairs; the energy terminology in our decision rule refers only to the inference-time transformation $E = -g_\theta$, not to a separate energy-model training loss.

## 4. Experimental Evaluation

In our experimental evaluation, we test STORM under an *annotation-free inference* setting, where no manual masks, boxes, or interactive prompts are provided at test time and no object-specific fine-tuning is performed on novel instances. We use "zero-shot" to denote *annotation-free and*

*Table 1.* STORM improves annotation-free estimation. We report AUC-ADD and AUC-ADD-S (Xiang et al., 2017) (*cf.* Appendix B; higher is better) and Average Recall (AR) on LM-O and YCB-Video. Results are reported as mean $\pm$ std over 5 independent evaluation runs (different random seeds). STORM approaches the performance obtained with ground-truth masks and outperforms the CNOS-based annotation-free baseline.

| | Method | $ADD_{AUC}$ | $ADD\text{-}S_{AUC}$ | $AR$ |
|---|---|---|---|---|
| **LM-O** | FP + CNOS | 57.0 | 68.0 | 41.0 |
| | **STORM (Ours)** | **74.0**$_{\pm1.28}$ | **89.0**$_{\pm1.25}$ | **53.0**$_{\pm2.02}$ |
| | FP + Ground Truth | 78.0 | 93.0 | 56.0 |
| **YCB-V** | FP + CNOS | 73.0 | 92.0 | 69.0 |
| | **STORM (Ours)** | **77.0**$_{\pm1.25}$ | **98.0**$_{\pm1.20}$ | **73.0**$_{\pm1.23}$ |
| | FP + Ground Truth | 78.0 | 99.0 | 74.0 |

*adaptation-free inference*: BOP test scenes and test images are not used for training, and no test-time masks, boxes, gradient updates, or object-specific weights are introduced. The same benchmark object identities may appear in BOP train and test splits, so this should not be read as category-disjoint novel-object generalization. We decompose the evaluation into five research questions that correspond to STORM's closed-loop pipeline:

**RQ1**: Can STORM achieve accurate **fully automatic 6D pose estimation** by coupling SOM masks with a fixed downstream pose estimator, approaching the upper bound given by ground-truth masks?

**RQ2**: Does SOM provide **high-quality and efficient reference-conditioned segmentation** under the BOP instance-segmentation protocol?

**RQ3**: Are SAM3D **reconstructions sufficiently** accurate for downstream pose registration, and how much does model alignment affect pose accuracy?

**RQ4**: Can TOM reliably **detect tracking drift** and enable closed-loop recovery, improving tracking robustness under occlusion and rapid motion with minimal overhead?

**RQ5**: How effective is the proposed attention-based Tracking Object Module (TOM) at **detecting tracking failures**?

Finally, we provide ablations that diagnose key design choices, including HSFA depth, language injection, and the effect of attention in TOM.

### 4.1. Fully Automatic 6D Pose Estimation

To answer **RQ1**, we evaluate STORM on challenging industrial datasets for fully automated 6D pose estimation. We used the LineMOD-Occluded (LM-O) (Brachmann et al., 2014) and YCB-Video (Xiang et al., 2017) established benchmarks. LM-O extends LineMOD, introducing significant inter-object occlusion, allowing to test robustness under cluttered and occluded scenarios. YCB-Video provides real-world video sequences of household objects with

6D pose annotations, ideal to evaluate pose estimation under varying viewpoints and lighting conditions.

We compare STORM against two reference points: (i) FoundationPose + CNOS (Nguyen et al., 2023), a strong annotation-free pipeline that predicts object masks via reference-conditioned matching, and (ii) FoundationPose + Ground-Truth masks, an oracle upper bound that isolates the impact of perfect segmentation on pose accuracy. All methods use the same downstream pose estimator, and only the mask source differs in each case.

We evaluated both datasets using standard metrics from the BOP benchmark: ADD, ADD-S (Xiang et al., 2017), and Average Recall (AR) (Hodaň et al., 2020). The ADD metric measures the average distance between corresponding 3D model points transformed by the predicted and ground-truth poses. ADD-S is a variant that handles pose ambiguities for objects that present symmetries by computing the distance to the nearest model point (*cf.* Appendix B for details).

**Results.** Figure 3 provides a qualitative comparison, visually confirming the accuracy of STORM's predictions. Table 1 summarizes the quantitative results. On the LM-O dataset, STORM outperforms all baseline methods and achieves performance within just 4% of the variant that uses ground-truth masks. On the YCB-Video dataset, STORM matches the ground-truth setup exactly and exceeds the strong CNOS baseline by 5%. STORM substantially narrows the gap to the manual-mask upper bound, underscoring the effectiveness of its annotation-free mask predictions.

### 4.2. Segmentation Quality and Efficiency

To answer **RQ2**, we evaluate the quality and the precision of the generated segmentation masks of the segmentation module (SOM) of STORM on several established benchmarks.

Our experiments are conducted on 5 core test datasets from the BOP benchmark suite (Hodaň et al., 2020): LM-O, T-LESS (Hodàn et al., 2017), TUD-Light (Hodaň et al., 2020), HomebrewedDB (Kaskman et al., 2019), and YCB-Video (Xiang et al., 2017). BOP (Benchmark for 6D Object Pose Estimation) is a unified framework for evaluating pose estimation systems under consistent protocols with occlusions, symmetries, and cluttered backgrounds. The selected benchmarks span 95 distinct object types, encompassing textured and textureless objects, symmetric and asymmetric shapes, and both household and industrial items, providing a diverse testbed for segmentation quality.

We compare STORM against the strongest publicly available systems on the BOP leaderboard: ZebraPoseSAT–EffNetB4 (Su et al., 2022), CosyPose (Labbé et al., 2020) and Mask R-CNN (He et al., 2017), which are fine-tuned per object category, as well as NOCTIS, LDSeg, MUSE, Prisma-MPG-Complex, NIDS, SAM6D (Lin et al., 2024),

*Table 2.* **STORM (SOM) achieves high-quality and efficient segmentation from a single reference image.** Comparison of annotation-free/adaptation-free and supervised segmentation methods on five BOP datasets. For STORM (SOM), AP scores are reported under the BOP target protocol with a single reference view ($k = 1$) and language conditioning; the mean column corresponds to $AP_C$. The $\pm$ values report five-seed variation for STORM (SOM); second-order metrics were not available for the baselines. The STORM runtime reports full SOM latency for the same $k = 1$ language-conditioned setting on an H100. Annotation-free methods, including STORM (SOM), operate without test-time masks, boxes, object-specific updates, or per-object retraining, whereas supervised baselines are fine-tuned per object category. STORM (SOM) achieves the highest mean AP while also being highly efficient in runtime. Within each method block, best values are in bold and $\circ$ depicts the second best. SOM stands for the Segmenting Object Module in the STORM architecture.

| | mAP ($\uparrow$) | LM-O | T-LESS | TUD-L | HB | YCB-V | Mean $\uparrow$ | Time (s) |
|---|---|---|---|---|---|---|---|---|
| *Annotation / adaptation-free* | **STORM (SOM, Ours)** | **57.8**$_{\pm2.18}$ | **53.0**$_{\pm1.67}$ | **73.3**$_{\pm2.26}$ | **74.1**$_{\pm0.50}$ | **80.3**$_{\pm1.41}$ | **67.7**$_{\pm1.44}$ | **0.046** |
| | NOCTIS | $\circ$48.9 | 47.9 | 58.3 | 60.7 | $\circ$68.4 | $\circ$56.8 | 0.990 |
| | LDSeg | 47.8 | $\circ$48.8 | $\circ$58.7 | $\circ$62.2 | 64.7 | 56.4 | 1.890 |
| | MUSE | 47.8 | 45.1 | 56.5 | 59.7 | 67.2 | 55.3 | 0.559 |
| | NIDS | 43.9 | $\circ$49.6 | 55.6 | 62.0 | 65.0 | 55.2 | 0.485 |
| | Prisma-MPG-Complex | 46.0 | 45.8 | 58.4 | 59.6 | 61.0 | 54.2 | 1.276 |
| | SAM6D | 46.0 | 45.1 | 56.9 | 59.3 | 60.5 | 53.6 | 2.795 |
| | ViewInvDet | 41.0 | 38.5 | 46.4 | 54.5 | 61.6 | 48.4 | 1.700 |
| | CNOS (FastSAM) | 39.7 | 37.4 | 48.0 | 51.1 | 59.9 | 47.2 | $\circ$0.221 |
| | CNOS (SAM) | 39.6 | 39.7 | 39.1 | 48.0 | 59.5 | 45.2 | 1.847 |
| *Supervised* | ZebraPoseSAT–EffNetB4 | **51.6** | **72.1** | **71.8** | **68.9** | **73.1** | **67.5** | $\circ$0.080 |
| | CosyPose | $\circ$37.5 | $\circ$54.4 | $\circ$48.9 | $\circ$47.1 | $\circ$52.0 | $\circ$48.0 | **0.050** |
| | Mask R-CNN | 37.5 | 54.4 | 48.9 | 47.1 | 42.9 | 46.2 | 0.100 |

CNOS, and ViewInvDet. Most baselines are distributed solely as executable packages via the BOP Challenge website without linked peer-reviewed publications, collectively referred to as the BOP Challenge Leaderboard[2]. STORM is evaluated under the same protocol for a fair comparison.

The final SOM release is trained once offline from BOP training splits (LM, LM-O, T-LESS, TUD-L, HB, and YCB-V) and auxiliary segmentation data from SA-V and RUAPC. BOP *test* scenes and images are never used for SOM training, and we do not fine-tune or adapt the model per object at evaluation time. At evaluation time on BOP, SOM can use between 1 and 16 rendered reference views per object, including the single-reference case, to condition HSFA for reference–query alignment. Unless otherwise stated, the main comparison in Table 2 uses a single reference view ($k = 1$) with language conditioning, matching the single-reference setting emphasized by STORM.

**Results.** Table 2 presents a comparison of segmentation methods across the BOP datasets. ZebraPoseSAT–EffNetB4 achieves a mean Average Precision (Lin et al., 2014) of 67.5%, making it the strongest supervised baseline. Our segmentation module SOM achieves a higher $AP_C$ of 67.7% under the BOP target protocol with a single reference view ($k = 1$) and language conditioning. Notably, SOM establishes a new state-of-the-art on four out of the five benchmarks—LM-O, TUD-L, HomebrewedDB, and YCB-

Video—and is the strongest annotation-free method on T-LESS, without any fine-tuning or retraining for specific object instances. These results highlight SOM's strong generalization ability and efficiency even in the single-reference setting. A component-level profile of the final release checkpoint is provided in Appendix E; with cached language descriptors, the single-reference setting used in Table 2 runs at about 22 FPS on an H100, while multi-reference settings provide a small accuracy–latency trade-off.

### 4.3. Impact of 3D Model Quality on Pose Estimation

To answer **RQ3**, we assess whether object models reconstructed from reference images are sufficient for downstream 6D pose estimation. Specifically, we evaluate on YCB-Video (YCB-V) under a controlled protocol that fixes the downstream pose estimator and the mask input, and, in turn, varies only the object model: (i) ground-truth CAD model, (ii) SAM3D reconstruction without alignment, and (iii) SAM3D reconstruction with our alignment procedure, as detailed in Appendix G.2.

**Results.** Table 3 shows that SAM3D reconstructions can provide useful geometry for downstream pose estimation when their coordinate frame is aligned to the benchmark convention. In the aligned setting, SAM3D+FP approaches GT+FP across VSDmean, IoUmean, and ADD-S$_{AUC}$, while still retaining a visible gap in IoU. These results support using SAM3D-reconstructed models as practical geometry for pose registration in our pipeline, but we do not treat

---

[2] https://bop.felk.cvut.cz/challenges/

*Table 3.* **SAM3D-generated 3D models provide reliable input for pose estimation.** Mean pose metrics (VSD, IoU, ADD-S AUC) on YCB-V for SAM3D (unaligned) + FoundationPose, SAM3D (aligned) + FoundationPose, and GT + FoundationPose. Results are reported as mean ± standard deviation over 5 independent end-to-end runs with different random seeds, where each run reruns the full pipeline from scratch (SAM3D reconstruction, optional alignment, and FoundationPose inference).

| Method | VSD↑ | IoU | ADD-S$_{AUC}$ |
|---|---|---|---|
| SAM3D (unaligned) + FP | 22.19$_{\pm 3.3}$ | 48.27$_{\pm 6.24}$ | 44.92$_{\pm 5.23}$ |
| SAM3D (aligned) + FP | 60.72$_{\pm 0.32}$ | 84.63$_{\pm 1.43}$ | 99.12$_{\pm 0.87}$ |
| GT + FP | 60.97 | 88.50 | 98.95 |

*Table 4.* **STORM automatically recovers from tracking failures.** TOM detects tracking loss via a persistent low-compatibility score and triggers re-tracking, while tracking fails to recover once drift accumulates. We use a *single global* threshold $\tau$ calibrated once on a held-out validation split of the synthetic verification dataset (Appendix H.1) and keep it fixed for all stress-test settings.

| Method | ADD$_{AUC}$ | ADD-S$_{AUC}$ | AR ↑ | Time(ms) |
|---|---|---|---|---|
| **STORM** | **74.64**$_{\pm 1.25}$ | **88.56**$_{\pm 1.89}$ | **67.85**$_{\pm 2.13}$ | 98$_{\pm 3}$ |
| FP Tracking | 52.76 | 66.76 | 50.09 | **84** |

*Table 5.* **STORM gains performance by attention-based matching.** We report Accuracy, AUROC, and per-frame inference time as the mean ± standard deviation over five independent runs.

| Models | Acc. (%) | AUROC (%) | Time (ms) |
|---|---|---|---|
| Cosine Similarity | 87.55 | 94.2 | 12.4 |
| TOM (w/o Attn.) | 95.23$_{\pm 0.26}$ | 95.6$_{\pm 0.16}$ | **12.2**$_{\pm 0.30}$ |
| TOM (Attn., 1L) | **98.36**$_{\pm 0.21}$ | **96.4**$_{\pm 0.13}$ | 13.5$_{\pm 0.37}$ |
| TOM (Attn., 2L) | 97.84$_{\pm 0.14}$ | 96.2$_{\pm 0.10}$ | 16.8$_{\pm 0.43}$ |

them as exact replacements for ground-truth CAD models. The CAD-based alignment is only used to report under the BOP coordinate convention; deployment requires a self-consistent reconstructed object frame rather than curated benchmark CAD assets.

### 4.4. Tracking Failure Detection and Recovery

To answer **RQ4**, we evaluate STORM's tracking module (TOM), *i.e.*, its ability to detect when the current observation is no longer compatible with previously successful tracks. Following our verifier formulation (§3.2), TOM predicts a compatibility logit $g_\theta(x_t, \mathcal{M})$ between the current observation $x_t$ and a memory pool $\mathcal{M}$, and converts it into an energy-like score $E(x_t, \mathcal{M}) \approx -g_\theta(x_t, \mathcal{M})$. Tracking loss is detected when the smoothed score persistently exceeds a calibrated threshold.

**Synthetic Failure Dataset from BOP.**  We construct a verification dataset using BOP annotations. For each object, we extract ground-truth masks to build a memory pool $\mathcal{M}$ of clean object crops from successful frames. For compatible samples $\mathcal{D}_{in}$, we pair a segmented observation with a crop from the *same* object, which should yield a high compatibility logit and thus a low energy-like score. Incompatible samples $\mathcal{D}_{out}$ are generated via: (i) *distractor swapping*, pairing observations with different object crops to simulate identity confusion; and (ii) *drift simulation*, using randomly shifted crops or randomly cropped regions to mimic tracking drift and background latch. We train TOM using binary cross-entropy over logits.

**Stress Test on Modified YCB-Video.**  To evaluate TOM under challenging conditions (rapid motion and occlusion), we further test on YCB-Video with controlled corruptions. We simulate high-speed motion by randomly dropping frames and simulate occlusion by applying random masks over object regions. These perturbations intentionally break temporal continuity, causing conventional trackers to drift without explicit failure detection.

Unless otherwise stated, all TOM hyperparameters (including $\tau$) are here fixed once after validation and are not further re-tuned for any corruption setting.

**Results.** Table 4 reports mean recall (ADD, ADD-S, AR) on the modified YCB-V dataset. With TOM-based tracking-loss detection, STORM significantly outperforms the baseline (*i.e.*without failure detection), with a measured end-to-end tracking-loop increase of about 14ms in this stress-test pipeline. The component profile measures the TOM verifier itself at 9.10$_{\pm 0.04}$ms per full forward pass (Appendix E); together with FoundationPose tracking, this keeps the steady-state tracking loop in the real-time regime. SOM is invoked only for initialization or drift-triggered re-localization, so we do not claim that all initialization and recovery stages run at the same per-frame rate.

This supports a key insight of our framework: once the observation becomes incompatible with the tracked-object memory, drift is often irrecoverable for standard temporal trackers, especially at longer sampling intervals in practice. Explicitly detecting persistent low-compatibility states enables timely reset and recovery accordingly. Figure 4 further illustrates this behavior clearly. Under camera rotation, both STORM and FoundationPose lose the target, but STORM identifies the failure and successfully re-tracks, whereas FoundationPose continues drifting and fails to recover due to the lack of a failure detection mechanism.

### 4.5. Tracking Loss Verification with Attention

To answer **RQ5**, we investigate how the matching design used to compute the compatibility logit $g_\theta(x_t, \mathcal{M})$ impacts tracking-loss detection. Since TOM uses $E(x_t, \mathcal{M}) \approx -g_\theta(x_t, \mathcal{M})$ only as an inference-time score, a stronger

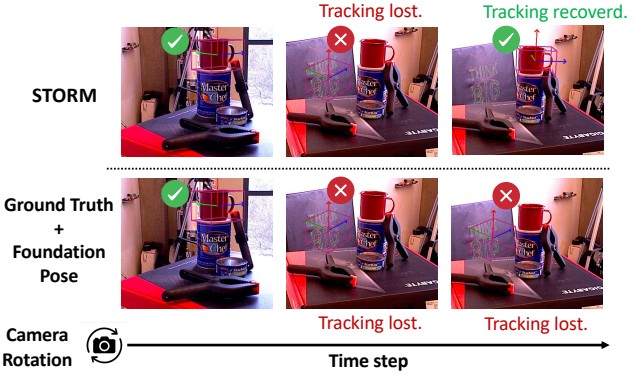

**STORM**

Tracking lost.

Tracking recoverd.

**Ground Truth + Foundation Pose**

Tracking lost.

Tracking lost.

**Camera Rotation**

**Time step**

*Figure 4.* **STORM automatically recovers from tracking failures.** A demonstration of the Tracking Object Module (TOM) successfully re-tracking a lost object. In contrast, FoundationPose fails to recover once tracking is lost. Best viewed in color.

matching mechanism should, in principle, increase the margin between compatible tracking states and failure states, thereby improving failure separability. Accordingly, using our synthetic failure dataset, we evaluate TOM under different matching configurations and compare it against a cosine-similarity baseline with a fixed metric.

**Results.** Table 5 summarizes tracking-loss detection performance and per-frame inference time across all matching variants. Introducing a single cross-attention layer between the current observation and the memory pool significantly improves failure-state separation, increasing accuracy from 87.55% to 98.36% with only 1.3ms additional latency, while also improving AUROC. Even without attention, TOM learns a discriminative compatibility function, achieving 95.23% accuracy while being the fastest (12.2ms) among learned models. This suggests that a learned logit provides a more reliable failure score than fixed feature matching, especially under distribution shift. Stacking a second attention layer does not yield further gains (97.84%) but instead increases latency to 16.8ms, indicating diminishing returns with deeper attention in practice. Overall, attention-based matching is important for reliable tracking-loss verification, and thus a single attention layer offers the best robustness/time efficiency trade-off.

## 5. Conclusion

We introduced STORM, a unified framework for annotation-free, reference-conditioned 6D pose tracking that addresses the **distributional shift** between canonical reference priors and dynamic observational queries. Instead of relying on template matching or curated CAD models at inference time, STORM combines **Hierarchical Spatial Fusion Attention**, which performs coarse-to-fine reference-query fusion with optional VLM semantics, and a **BCE-trained Tracking Object Module**, which converts a learned compatibility logit

into an energy-like score for temporal drift detection. This allows the system to segment objects under occlusion and to trigger re-initialization when tracking loss is detected. In our experiments, STORM achieves state-of-the-art performance on benchmarks, closing the gap between annotation-free inference and fully supervised baselines while incurring modest computational overhead. Beyond improving pose accuracy, the results highlight the value of explicitly modeling the reliability of the tracking state. SOM provides robust object masks from minimal reference information, while TOM gives the tracker a continuous self-checking signal that can distinguish transient uncertainty from persistent drift. This closed-loop design is particularly useful in dynamic scenes, where occlusion, fast motion, and viewpoint changes are common. The SAM3D analysis further suggests that reconstructed geometry can be sufficient for practical pose registration when coordinate consistency is handled carefully. Overall, STORM shows that segmentation, geometry, and failure awareness should be considered jointly in reference-conditioned pose tracking.

## 6. Limitations and Future Work

STORM inherits several boundaries from reference-conditioned perception and tracking. Although it can operate from one reference image, this is only the minimum input: viewpoint-diverse references improve robustness, and symmetric or textureless objects such as T-LESS can remain ambiguous for both visual matching and language descriptions. The full pipeline also depends on the fidelity and coordinate consistency of SAM3D reconstructions and on the frozen downstream FoundationPose registration/tracking backend; failures in these components can bottleneck pose quality, especially for transparent, reflective, or thin objects not extensively covered by our evaluation. Language conditioning is optional, and the VLM/CLIP descriptor is generated once per object and cached, but its benefit depends on descriptor quality and is most visible at low reference counts or in ambiguous scenes. TOM uses a validation-calibrated threshold and an $L$-consecutive-frame rule; while this reduces false alarms in our stress tests, online threshold adaptation for dynamic deployments remains an important future direction. Finally, our zero-shot setting denotes annotation-free and adaptation-free inference without test-time masks, boxes, or object-specific updates, rather than category-level novel-object generalization under a disjoint object split.

Future work will focus on making STORM adaptive and deployable. Promising directions include online calibration of TOM's failure threshold, stronger reconstruction and alignment for transparent, reflective, thin, or symmetric objects, richer semantic conditioning for visually similar instances, and evaluation on category-disjoint object splits, multi-object tracking, and robotic manipulation tasks.

## Acknowledgements

This work was supported by the European Union (Grant No. 57100467), the Federal Ministry of Research, Technology and Space within the JUPITER AI Factory, and the Federal Ministry of Food and Agriculture within the FarmerSpaceAI project (Grant No. 28DE401E23).

## Impact Statement

This work aims to advance computer vision by providing an annotation-free inference framework for 6D pose tracking, significantly reducing the labor required to deploy physical AI systems. By integrating explicit tracking-loss detection, STORM improves the reliability and safety of autonomous agents in dynamic environments like logistics and manufacturing. While these advancements facilitate broader robotic accessibility, we do not foresee specific negative societal consequences warranting individual highlighting here.

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

## A. HSFA Algorithm

Algorithm 1 outlines our proposed HSFA mechanism, designed to integrate multimodal features and exploit spatial interactions across query and reference views in a structured, iterative fashion. At the core of the algorithm lies a multi-stage attention pipeline operating over $n$ iterative fusion steps.

In each iteration, language conditioning is applied as zero-initialized shift/scale modulation before each query sub-block, while a separate language gate modulates the reference key/value channels in cross-attention. The query features $E^q$ are first updated via self-attention to enhance internal dependencies, then fused with gated reference features using cross-attention, and finally refined by an MLP.

Next, the algorithm processes the reference features by first decomposing the concatenated $E^{\text{ref}}$ into a set of $m$ individual reference features $E_1^r, \ldots, E_m^r$, each corresponding to a separate reference image (Line 8). Each of these reference feature maps is independently refined via self-attention (Line 10), capturing local spatial patterns within each view. The updated features are then reassembled into a single tensor (Line 12) and subjected to a final round of global self-attention (Line 14), modeling inter-image relations across the entire reference set.

After $n$ fusion iterations, the refined query representation is passed through an upsampling convolutional block to generate a high-resolution prediction (Line 17). This final output, denoted as $H$, forms the region of interest prediction with enhanced spatial and multimodal coherence.

## B. Metric Details

### B.1. ADD (Average Distance of Model Points)

The ADD metric measures the average Euclidean distance between the 3D model points transformed by the predicted pose and those transformed by the ground-truth pose. It is defined as:

$$\text{ADD} = \frac{1}{|M|} \sum_{x \in M} \|(Rx + t) - (R_{\text{gt}}x + t_{\text{gt}})\|, \quad (4)$$

where $M$ is the set of 3D model points, $(R, t)$ is the estimated rotation and translation, and $(R_{\text{gt}}, t_{\text{gt}})$ is the ground-truth pose. A prediction is considered correct if the ADD score is below a certain threshold, typically set to 10% of the object's diameter.

### B.2. ADD-S (Average Distance for Symmetric Objects)

For symmetric objects, the ADD-S metric is used to account for pose ambiguity. Instead of comparing each model point

---

**Algorithm 1** Hierarchical Spatial Fusion Attention

1: **for** $j = 1$ **to** $n$ **do**
2:     *## Query self-attention with language-conditioned normalization*
3:     $\tilde{E}^q \leftarrow \text{AdaFiLM}_0(E^q, E^t)$
4:     $E^q \leftarrow E^q + \text{SelfAttention}(\tilde{E}^q)$
5:     *## Query-reference fusion with language-gated reference channels*
6:     $\tilde{E}^q \leftarrow \text{AdaFiLM}_0(E^q, E^t)$
7:     $\tilde{E}^{\text{ref}} \leftarrow \text{LangGate}(E^{\text{ref}}, E^t)$
8:     $E^q \leftarrow E^q + \text{CrossAttention}(\tilde{E}^q, \tilde{E}^{\text{ref}})$
9:     *## Query refinement*
10:     $\tilde{E}^q \leftarrow \text{AdaFiLM}_0(E^q, E^t)$
11:     $E^q \leftarrow E^q + \text{MLP}(\tilde{E}^q)$
12:     $\{E_1^r, \ldots, E_m^r\} \leftarrow \text{Split}(E^{\text{ref}})$
13:     *## Split the concatenated reference features back into individual image features*
14:     **for** $i = 1$ **to** $m$ **do**
15:         $E_i^r \leftarrow \text{SelfAttention}(E_i^r)$
16:         *## Update each reference image feature independently with self-attention*
17:     **end for**
18:     $E^{\text{ref}} \leftarrow \text{Concat}(E_1^r, \ldots, E_m^r)$
19:     *## Re-concatenate the updated reference features*
20:     $E^{\text{ref}} \leftarrow \text{SelfAttention}(E^{\text{ref}})$
21:     *## Apply self-attention on the concatenated reference features to model global interactions*
22: **end for**
23: $H \leftarrow \text{UpsampleConvBlock}(E^q)$
24: *## Upsample and refine query features through an upsampling convolutional block*
25: *## Return the final high-resolution region of interest*

---

to its exact correspondence, ADD-S computes the distance to the nearest model point under the predicted and ground-truth poses:

$$\text{ADD-S} = \frac{1}{|M|} \sum_{x_1 \in M} \min_{x_2 \in M} \|(Rx_1 + t) - (R_{\text{gt}}x_2 + t_{\text{gt}})\|. \quad (5)$$

This formulation handles symmetrical cases (e.g., cylindrical or handle-less objects) where multiple poses result in the same visual appearance.

### B.2.1. Interpretation and AUC Reporting

ADD and ADD-S are distance errors: smaller values indicate better pose alignment. Throughout this paper, however, we report their AUC variants, where larger values are better.

Given per-instance distances $\{d_i\}_{i=1}^N$ (using ADD for asymmetric objects and ADD-S for symmetric ones), we define

an accuracy–threshold curve:

$$\mathrm{Acc}(\tau) = \frac{1}{N} \sum_{i=1}^{N} \mathbb{I}\left[d_i < \tau\right]. \qquad (6)$$

We compute the area under this curve up to a maximum threshold $\tau_{\max}$ and normalize it:

$$\mathrm{AUC} = \frac{1}{\tau_{\max}} \int_0^{\tau_{\max}} \mathrm{Acc}(\tau)\, d\tau, \quad \tau_{\max} = 0.1\, d, \quad (7)$$

where $d$ denotes the object diameter. In practice, we approximate the integral with $K$ uniformly spaced thresholds $\tau_k = \frac{k}{K}\tau_{\max}$:

$$\mathrm{AUC} \approx \frac{1}{K} \sum_{k=1}^{K} \mathrm{Acc}(\tau_k). \qquad (8)$$

We denote these scores as $\mathrm{ADD}_{\mathrm{AUC}}$ and $\mathrm{ADD\text{-}S}_{\mathrm{AUC}}$. In the paper, we report all AUC values in percentages, i.e., $100 \times \mathrm{ADD}_{\mathrm{AUC}}$ and $100 \times \mathrm{ADD\text{-}S}_{\mathrm{AUC}}$ (higher is better).

## B.3. mAP (Mean Average Precision)

For BOP dataset evaluation, we exclude all test samples with visibility lower than 0.1 and evaluate it by mAP, the overall score is computed in three stages.

### B.3.1. PER-CATEGORY AP

Define the set of IoU thresholds

$$T = \{0.50, 0.55, \ldots, 0.95\}. \qquad (9)$$

For each category $c$,

$$\mathrm{AP}_c = \frac{1}{|T|} \sum_{t \in T} \mathrm{Precision}_c(\mathrm{IoU} = t). \qquad (10)$$

Only instances with visible area $\geq 10\%$ are counted; smaller ones are ignored.

### B.3.2. DATASET-LEVEL MEAN AP

For each dataset $d$ with category set $\mathcal{C}_d$,

$$\mathrm{AP}_d = \frac{1}{|\mathcal{C}_d|} \sum_{c \in \mathcal{C}_d} \mathrm{AP}_c. \qquad (11)$$

At evaluation time, only the top-100 predictions per image are kept.

### B.3.3. OVERALL MAP

Let the core datasets be $\mathcal{D}$.

$$\mathrm{mAP} = \frac{1}{|\mathcal{D}|} \sum_{d \in \mathcal{D}} \mathrm{AP}_d. \qquad (12)$$

## B.4. AR (Average Recall)

### B.4.1. PER-ERROR-FUNCTION AVERAGE RECALL

An estimated pose is correct w.r.t. error function $e$ if $e < \theta_e$. Define the set of thresholds $\Theta_e$ (and misalignment tolerances $\tau$ for VSD):

$$AR_e = \frac{1}{|\Theta_e|} \sum_{(\theta,\tau) \in \Theta_e} \mathrm{Recall}_e(\theta, \tau). \qquad (13)$$

### B.4.2. DATASET-LEVEL AVERAGE RECALL

For dataset $d$,

$$AR_d = \frac{1}{3}\left(AR_{\mathrm{VSD}} + AR_{\mathrm{MSSD}} + AR_{\mathrm{MSPD}}\right). \qquad (14)$$

**Overall Average Recall.** Let the core datasets be $\mathcal{D}$. Then

$$AR_C = \frac{1}{|\mathcal{D}|} \sum_{d \in \mathcal{D}} AR_d. \qquad (15)$$

## B.5. AUROC (Area Under the ROC Curve)

In tracking-loss verification, TOM outputs an energy-like score $E(x, M)$ (or its EMA-smoothed version $\tilde{E}$), where a *higher* score indicates a higher likelihood of tracking failure. We treat incompatible/failure pairs $D_{\mathrm{out}}$ as the positive class and compatible pairs $D_{\mathrm{in}}$ as the negative class. Given a threshold $\tau$, we predict LOST if $E(x, M) > \tau$.

### B.5.1. ROC CURVE

Sweeping $\tau$ yields the receiver operating characteristic (ROC) curve, defined by the true positive rate (TPR) and false positive rate (FPR):

$$\mathrm{TPR}(\tau) = \frac{|\{(x, M) \in D_{\mathrm{out}} : E(x, M) > \tau\}|}{|D_{\mathrm{out}}|}, \quad (16)$$

$$\mathrm{FPR}(\tau) = \frac{|\{(x, M) \in D_{\mathrm{in}} : E(x, M) > \tau\}|}{|D_{\mathrm{in}}|}. \quad (17)$$

### B.5.2. AUROC

The area under the ROC curve (AUROC) summarizes performance across all thresholds:

$$\mathrm{AUROC} = \int_0^1 \mathrm{TPR}\left(\mathrm{FPR}^{-1}(u)\right) du. \qquad (18)$$

Equivalently, AUROC can be interpreted as the probability that a randomly sampled failure pair receives a higher score than a randomly sampled compatible pair (ties counted as 0.5):

$$\mathrm{AUROC} = \Pr\left(E(x^+, M^+) > E(x^-, M^-)\right). \quad (19)$$

AUROC ranges from 0 to 1, where 0.5 corresponds to random ranking and 1.0 indicates perfect separability.

## C. Experimental Details

### C.1. Hardware Information

Unless otherwise stated, training experiments are conducted on a server equipped with an AMD EPYC 7343 16-Core Processor (2.0–3.2 GHz, 64 threads), 500 GB of DDR4 RAM with an additional 8 GB swap space, and **eight NVIDIA A100 GPUs** (each with 80 GB of HBM2e memory). Runtime profiling is reported separately on an NVIDIA H100 GPU in Appendix E.

### C.2. Training Details

The final expanded SOM model is initialized from the base SOM model and further trained on an expanded robustness mix. The training data consist of the *training splits* of LM (Hinterstoisser et al., 2012), LMO, YCBV, HB, TLESS, and TUDL, together with RUAPC (Rennie et al., 2016) and the SA-V (Segment Anything Video) dataset released with SAM 2 (Ravi et al., 2024). BOP test scenes and test images are not used for training. All SOM training is conducted on eight NVIDIA A100 GPUs (80GB each).

To enhance robustness, we use photometric and geometric augmentations including lighting changes, rotations, affine transforms, and random erasing. The learning-rate schedule uses a warm-up phase followed by cosine annealing (Loshchilov & Hutter, 2016), with gradient clipping (Pascanu et al., 2013) to mitigate instability. SOM uses frozen DINOv3 visual encoders with lightweight trainable adaptation and task-specific fusion/decoder modules. The test set is drawn from the BOP Challenge test protocol.

**Reference Construction and Multi-View Sampling.** For SA-V, we build the reference set for each object by collecting multiple *segmented object images* (foreground crops) from different frames. During training, the number of references used per instance is randomly sampled as $k \in \{1, \ldots, 16\}$ with a uniform distribution, such that instances with a single reference and those with multiple references appear with equal probability.

For datasets with available 3D object models (e.g., BOP datasets), we additionally render each object model from 16 uniformly sampled spherical viewpoints to form a synthetic reference pool. During training, we use both (i) rendered reference images and (ii) segmented object images as references, and similarly sample $k \in \{1, \ldots, 16\}$ uniformly for each instance. Subsequently, HSFA is applied to align features from the sampled reference views and facilitate interaction with query views, effectively suppressing activations in background regions. Notably, rendered reference views are only a convenient way to increase view diversity when a 3D model is available. They are not required by the method: in our pipeline, the same rendered-view conditioning can be obtained by rendering from reconstructed object meshes (e.g., produced from reference images), without relying on curated CAD assets.

## D. Implementation Details

### D.1. Implementation of SOM

#### D.1.1. DATA AUGMENTATION

During training, each query image and its corresponding mask are subjected to joint photometric and geometric transforms. Specifically:

**Photometric Augmentation.** We apply ColorJitter with brightness variation of $\pm 30\%$, contrast variation of $\pm 30\%$, saturation variation of $\pm 30\%$, and hue adjustment of $\pm 0.1$.

**Geometric Augmentation.** A horizontal flip is performed with probability 0.5, synchronized between the image and its mask. In addition, a random affine transform is applied with rotation uniformly sampled from $[-180°, +180°]$, scale sampled from $[0.95, 1.05]$, and shear sampled from $[-25°, +25°]$; bicubic interpolation is used for images, while nearest-neighbor interpolation is used for masks.

**Resize & Crop.** All outputs are first resized so that the shorter edge matches the target length (using bicubic interpolation for images and nearest-neighbor for masks), then center-cropped to a spatial size of $H \times W$.

**Tensor Conversion & Normalization.** Images are converted to tensors and normalized using ImageNet statistics $\mu = [0.485, 0.456, 0.406]$ and $\sigma = [0.229, 0.224, 0.225]$; masks are converted to single-channel tensors without further normalization.

**Random Erasing.** With probability $p = 0.5$, we erase a random rectangle covering $2-10\%$ of the image area (aspect ratio between 0.3 and 3.3), filling it with random pixel values in the image and zeros in the mask.

**VLM Prompt for Semantic Prior.** As described in Sec. 3.1.1, we query a vision-language model with the reference image $I_{\mathrm{ref}}$ and a generic prompt $p$ to obtain a concise object descriptor $T$, which is then encoded by a frozen CLIP text encoder and injected into the visual backbone via zero-initialized AdaLN/FiLM-style conditioning. The descriptor is object-level and can be cached once references are fixed, so language conditioning does not require a new VLM call for every video frame.

We use the same prompt for all datasets:

```
Describe the main object in
```

```
the image with a short noun
phrase (at most 6 words).  Do
not mention background, colors,
materials, or the word ``image''.
Output only the phrase.
```

We post-process the VLM output by keeping the first line, stripping punctuation and surrounding whitespace, and low-ercasing. If the output is empty, we fall back to the generic token "object".

### D.1.2. TRAINING DETAILS

**Optimization & Learning Rate Schedule.** We optimize SOM with AdamW. For the expanded release training stage, the base learning rate is $1 \times 10^{-4}$, the context/refinement learning rate is $5 \times 10^{-4}$, and the LoRA learning rate is $2.5 \times 10^{-5}$. We use weight decay 0.1 for regular trainable parameters and LoRA weight decay 0.01. The global batch size is 256, training uses bf16 mixed precision, and the learning-rate schedule consists of a warm-up phase followed by cosine annealing. We apply gradient-norm clipping with maximum norm 1.0 after unscaling.

**Loss Function Hyperparameters.** Let $p_i \in [0, 1]$ denote the predicted foreground probability for pixel $i$ and $y_i \in \{0, 1\}$ the ground-truth label. We optimize a weighted sum of losses:

$$\mathcal{L}_{\text{total}} = 20.0\,\mathcal{L}_{\text{Focal}} + 1.0\,\mathcal{L}_{\text{Dice}} \\ + 1.0\,\mathcal{L}_{\text{IoUPred}} + 1.0\,\mathcal{L}_{\text{Det}}. \tag{20}$$

The implementation also supports IoU, Tversky, F-beta, boundary, matching, and deep-supervision losses, but these terms are inactive in the final SOM configuration.

**Focal Loss.** We use the binary focal loss:

$$\mathcal{L}_{\text{Focal}} = -\frac{1}{N} \sum_{i=1}^{N} \Big( \alpha\, y_i (1 - p_i)^{\gamma} \log(p_i) \\ + (1 - \alpha)\,(1 - y_i)\, p_i^{\gamma} \log(1 - p_i) \Big). \tag{21}$$

with $w_{\text{Focal}} = 20.0$, $\alpha = 0.25$, and $\gamma = 2.0$.

**Dice Loss.** We use the soft Dice loss:

$$\mathcal{L}_{\text{Dice}} = 1 - \frac{2 \sum_i p_i y_i + \epsilon}{\sum_i p_i + \sum_i y_i + \epsilon}, \tag{22}$$

with $w_{\text{Dice}} = 1.0$ and $\epsilon = 10^{-6}$.

**IoU-Prediction and Detection Auxiliary Terms.** The IoU-prediction term supervises the decoder's predicted mask-quality score with the thresholded mask IoU using an $\ell_1$ loss. The detection auxiliary term applies binary cross-entropy to the decoder's spatial detection logits against the ground-truth foreground mask. These two terms stabilize the mask-quality head and detection gate without adding any test-time annotation.

### D.2. Implementation of TOM

**Dataset** See Appendix H.3.

**Training Details** TOM takes as input an observation crop $x_t$ and a memory crop $M_t$. Both crops are first encoded by a frozen DINOv3 encoder to obtain feature maps, and TOM predicts a scalar logit $z$ from the resulting feature pair. We optimize TOM using binary cross-entropy with logits (BCEWithLogitsLoss) and train for 10 epochs with the Adam optimizer.

Training is conducted on a single NVIDIA A100 GPU (80GB) with a batch size of 64 crop pairs and a learning rate of $1 \times 10^{-4}$. We use Adam with default parameters $(\beta_1, \beta_2, \epsilon)$ as in PyTorch.

The BCEWithLogitsLoss is:

$$\mathcal{L}(\mathbf{z}, \mathbf{y}) = -\frac{1}{N} \sum_{i=1}^{N} \big[ y_i \log \sigma(z_i) + (1 - y_i) \log\big(1 - \sigma(z_i)\big) \big], \tag{23}$$

where $z_i$ is the predicted logit for the $i$-th crop pair, $y_i \in \{0, 1\}$ is the target label, $\sigma(\cdot)$ denotes the sigmoid function, and $N$ is the batch size.

## E. Runtime Profile

We profile the final SOM architecture on a single NVIDIA H100 80GB GPU using PyTorch 2.5.1, CUDA 12.1, bf16 precision, 512×512 SOM inputs, 3 warmup runs, and 10 timed repeats. Table 6 reports mean latency, standard deviation, and throughput. The VLM-generated object descriptor is object-level and can be cached once the references are fixed; the per-frame language cost in SOM is therefore the CLIP/text-conditioning path, not a repeated VLM call.

These numbers separate one-time initialization/recovery costs from steady-state tracking costs. During steady-state operation, STORM runs FoundationPose tracking and TOM drift detection per frame; TOM adds 9.10ms and the combined loop is approximately 25–33 FPS in our measurements. SOM is called during initialization or drift-triggered re-localization, where the release checkpoint runs at 22.8 FPS for $k = 1$, 21.2 FPS for $k = 8$, and 16.2 FPS for $k = 16$ with language conditioning. Heavier components such as VLM descriptor generation, SAM3D reconstruction, and FoundationPose registration are not per-frame costs in the steady-state tracking loop.

*Table 6.* **Release-checkpoint runtime profile.** SOM latency is measured at 512×512 with the final release checkpoint and bf16 precision. Each SOM cell reports latency in ms followed by throughput in FPS. TOM latency is measured for a full verifier forward pass at 224×224. Values are mean ± standard deviation over 10 timed repeats after warmup.

| Component | $k = 1$ ms (FPS) | $k = 4$ ms (FPS) | $k = 8$ ms (FPS) | $k = 16$ ms (FPS) |
|---|---|---|---|---|
| CLIP text encoder | 3.05±0.02 (328.3) | 3.05±0.02 (328.3) | 3.05±0.02 (328.3) | 3.05±0.02 (328.3) |
| DINOv3 query encoder | 12.62±0.04 (79.2) | 12.62±0.04 (79.2) | 12.62±0.04 (79.2) | 12.62±0.04 (79.2) |
| DINOv3 reference encoder | 12.10±0.04 (82.7) | 12.70±0.04 (78.8) | 15.82±0.05 (63.2) | 23.27±0.06 (43.0) |
| HSFA fusion, no language | 15.03±0.05 (66.5) | 15.23±0.05 (65.7) | 16.40±0.01 (61.0) | 23.06±0.04 (43.4) |
| HSFA fusion, language | 17.70±0.03 (56.5) | 17.88±0.04 (55.9) | 18.22±0.01 (54.9) | 24.97±0.04 (40.0) |
| Mask decoder, no language | 1.94±0.05 (516.1) | 1.94±0.05 (516.1) | 1.94±0.05 (516.1) | 1.94±0.05 (516.1) |
| Mask decoder, language | 1.88±0.02 (533.0) | 1.88±0.02 (533.0) | 1.88±0.02 (533.0) | 1.88±0.02 (533.0) |
| Full SOM, no language | 41.35±0.05 (24.2) | 41.96±0.07 (23.8) | 45.54±0.05 (22.0) | 59.76±0.18 (16.7) |
| Full SOM, language | 43.87±0.04 (22.8) | 44.49±0.08 (22.5) | 47.18±0.06 (21.2) | 61.65±0.04 (16.2) |

| TOM component | Latency ms (FPS) |
|---|---|
| Encode one crop | 4.71±0.03 (212.2) |
| Full TOM forward | 9.10±0.04 (109.9) |

# F. Ablation Details

## F.1. Settings.

To validate the effectiveness of SOM, we conducted the following experiments. All training was performed on eight A100 GPUs. Except for disabling all data augmentation, the remaining settings were consistent with those described in Appendix D. We evaluated loss convergence, convergence speed, and AP on the test dataset. Additionally, we examined the impact of language injection and the effects of using different HSFA layers.

## F.2. Reference Count and Language Conditioning in SOM

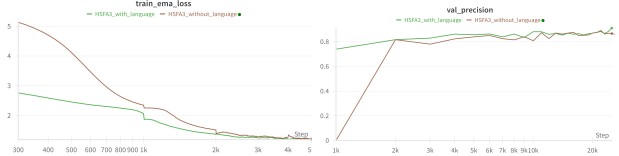

*Figure 5.* Under the same experimental conditions, we compared the convergence behavior of the training EMA loss and the prediction precision in test datasets for the HSFA3 layer, with and without language injection.

We evaluate how SOM responds to the number of reference views and to optional language conditioning under the same BOP target protocol used in Table 2. For $k < 16$, reference views are sampled with a fixed random seed; for $k = 16$, all available reference views are used. Table 7 shows that SOM already performs strongly with a single reference image,

improves with additional reference diversity, and largely saturates after a few views. Language conditioning provides a small but consistent gain, most visibly on LM-O and T-LESS where object appearance is more ambiguous. This supports the single-reference setting used in Table 2, while showing that additional reference views can provide small accuracy gains when available.

## F.3. Ablation on HSFA Module in SOM

As shown in Fig. 6, we compared the training loss, EMA loss, and AP on the test set for various HSFA layers. HSFA0 corresponds to the use of only simple attention, convolution, and linear layers. It can be observed that the HSFA0 model hardly converges to a satisfactory value, and the loss decreases very slowly. For HSFA3 and HSFA7, the convergence speed of HSFA7 is noticeably faster than that of HSFA3. However, the final results are not significantly different between the two. Therefore, in consideration of the trade-off between performance and speed, HSFA3 is preferred.

Table 8 summarizes the final results. It can be seen that HSFA7 achieves the best training loss, reaching 0.82, and also obtains the highest AP on the test set at 95.4%, which is one percentage point higher than HSFA3 and more than 50 percentage points higher than HSFA0.

## F.4. Ablation on DINO Module in SOM

The experimental settings remained the same as before. In this module, we primarily compared the impact of fine-tuning DinoV3 on our proposed framework. As shown in

*Table 7.* **SOM sensitivity to reference count and language conditioning.** Results are reported under the BOP target protocol without test-time augmentation. Per-dataset columns report $AP_D$; $AP_C$ is averaged over the five datasets. Time reports full SOM latency at 512×512 on an H100 with bf16 precision. For $k < 16$, reference views are sampled with a fixed seed; $k = 16$ uses all reference views.

| $k$ | Variant | LM-O | T-LESS | TUD-L | HB | YCB-V | $AP_C \uparrow$ | Time (ms) $\downarrow$ |
|---|---|---|---|---|---|---|---|---|
| 1 | SOM w/o lang | 57.1 | 52.1 | 73.1 | 74.1 | 80.1 | 67.3 | 42.8 |
| | SOM w/ lang | 57.8 | 53.0 | 73.3 | 74.1 | 80.3 | 67.7 | 45.8 |
| 2 | SOM w/o lang | 57.5 | 53.1 | 73.4 | 74.3 | 80.4 | 67.7 | 43.8 |
| | SOM w/ lang | 58.1 | 53.7 | 73.4 | 74.2 | 80.5 | 68.0 | 47.2 |
| 4 | SOM w/o lang | 57.9 | 53.5 | 73.2 | 74.2 | 80.4 | 67.8 | 43.5 |
| | SOM w/ lang | 58.4 | 53.9 | 73.3 | 74.2 | 80.5 | 68.1 | 46.4 |
| 8 | SOM w/o lang | 58.0 | 53.7 | 73.2 | 74.2 | 80.4 | 67.9 | 45.8 |
| | SOM w/ lang | 58.2 | 53.8 | 73.4 | 74.2 | 80.5 | 68.0 | 47.7 |
| 16 | SOM w/o lang | 58.0 | 53.8 | 73.2 | 74.3 | 80.4 | 67.9 | 60.3 |
| | SOM w/ lang | 58.2 | 53.9 | 73.3 | 74.2 | 80.5 | 68.0 | 61.8 |

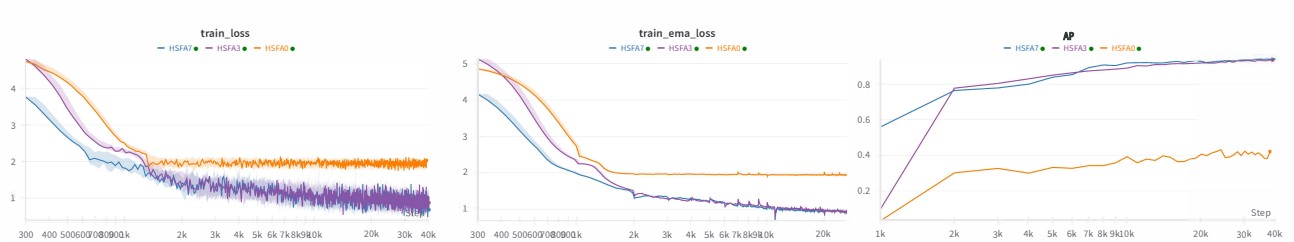

*Figure 6.* To provide a comprehensive comparison, we evaluated the training loss, training EMA loss, and test set AP for HSFA0, HSFA3, and HSFA7 layers under the same experimental conditions.

*Table 8.* SOM with different numbers of HSFA layers under the same BOP training protocol.

| Models | Training Loss | AP (%) |
|---|---|---|
| 0 layer HSFA | 1.96 | 44.0 |
| 3 layer HSFA | 0.86 | 94.6 |
| 7 layer HSFA | **0.82** | **95.4** |

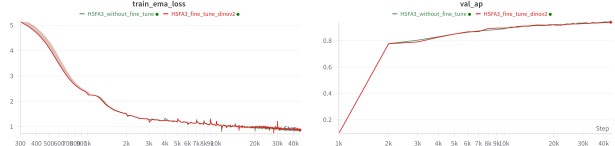

*Figure 7.* We conducted an ablation study to compare the effects of fine-tuning DinoV3 versus not fine-tuning it. Specifically, we compared the training EMA loss and the AP on the test set under both conditions.

Fig. 7, fine-tuning DinoV3 does not provide any substantial improvement to the overall performance of our architecture. Both the convergence speed and model robustness remain consistent regardless of whether fine-tuning is applied. Therefore, fine-tuning DinoV3 is not a critical factor in our system.

## G. SAM3D Extraction

### G.1. Model Reconstruction

This section illustrates SAM3D-based 3D model reconstruction from reference images. We use SAM3D to generate 3D models from reference images and align them with the provided CAD models for benchmark-coordinate evaluation.

Figure 8 provides a qualitative comparison between the ground-truth CAD models and the aligned 3D models reconstructed from reference images using SAM3D. Across diverse object types, including bottles and cans, the SAM3D-generated models capture the global geometry and many characteristic shape details of the ground-truth models, including object proportions, aspect ratios, bottle necks, rims, and can lids. The reconstructed meshes often exhibit smoother and more regular contours along object boundaries, which can be beneficial for downstream pose estimation and rendering robustness. At the same time, local surface detail, texture, and benchmark-coordinate alignment can differ from the provided CAD models, so we treat

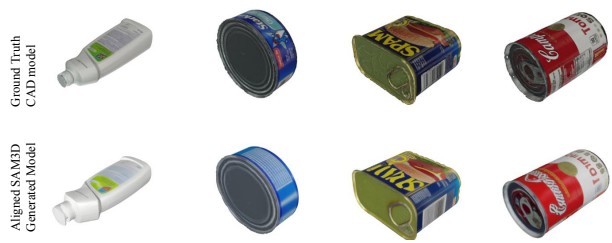

Ground Truth CAD model

Aligned SAM3D Generated Model

*Figure 8.* **Comparison of 3D models reconstructed from reference images and ground-truth 3D models.** The aligned SAM3D models recover the main object structure while producing smoother and more regular contours along object boundaries.

SAM3D reconstructions as practical geometry for downstream registration rather than as exact CAD replacements.

### G.2. Alignment of Models

SAM3D produces reconstructed meshes whose coordinate system may not match the BOP CAD convention, most notably in global scale and a rigid offset. To enable fair evaluation under BOP-style benchmarks—where object models are assumed to follow the provided CAD coordinate system and scale—we align each SAM3D mesh $\mathcal{M}_s$ to the corresponding benchmark CAD mesh $\mathcal{M}_c$ by estimating a similarity transform (isotropic scale + rotation + translation, without reflection). Importantly, this CAD-based alignment is *only* used to match the benchmark coordinate convention (and to report results under that convention); it is *not required* for inference or deployment in real robotic tasks. In practical robot setups, the reconstructed mesh can be used directly as the object model for pose estimation (possibly after coarse normalization such as centering/scaling), since the downstream estimator only requires a self-consistent object coordinate frame rather than alignment to an external CAD frame.

Given a pair $(\mathcal{M}_s, \mathcal{M}_c)$, we uniformly sample two point sets from their surfaces: $S = \{\mathbf{s}_i\}_{i=1}^N$ from $\mathcal{M}_s$ and $C = \{\mathbf{c}_j\}_{j=1}^N$ from $\mathcal{M}_c$ (we use $N = 20000$ in all experiments unless stated otherwise). We run ICP initialized with the identity transform. At each iteration, we compute nearest-neighbor correspondences from the transformed points in $S$ to $C$, and update the similarity transform parameters $(s, R, \mathbf{t})$ by minimizing the least-squares error under the current correspondences:

$$\min_{s>0,\ R\in SO(3),\ \mathbf{t}\in\mathbb{R}^3} \sum_{i=1}^N \big\| s R \mathbf{s}_i + \mathbf{t} \ - \ \pi_C\big(sR\mathbf{s}_i + \mathbf{t}\big) \big\|_2^2, \tag{24}$$

where $\pi_C(\cdot)$ denotes the nearest-neighbor operator in $C$. We repeat correspondence estimation and parameter update until convergence, then apply the final transform to all vertices of $\mathcal{M}_s$ to obtain the aligned mesh $\tilde{\mathcal{M}}_s$.

The alignment is performed during mesh preparation, after SAM3D mesh reconstruction. The resulting aligned mesh $\tilde{\mathcal{M}}_s$ is used as the object model for the downstream pose estimator (e.g., FoundationPose).

## H. TOM

### H.1. Energy-Like Decision Rule and Threshold Calibration

Given the compatibility logit $g_\theta(x_t, \mathcal{M}_t)$, we define the energy-like score as

$$E_t \triangleq E(x_t, \mathcal{M}_t) := -g_\theta(x_t, \mathcal{M}_t), \tag{25}$$

where higher $E_t$ indicates lower compatibility and thus a higher likelihood of tracking drift.

**Temporal Smoothing.** To avoid spurious spikes caused by motion blur or transient occlusions, we apply an exponential moving average (EMA):

$$\tilde{E}_t = \alpha\tilde{E}_{t-1} + (1-\alpha)E_t, \qquad \alpha \in [0,1). \tag{26}$$

In all experiments, we use a fixed EMA coefficient $\alpha = 0.9$.

**Loss Detection Rule.** We declare tracking loss if the smoothed energy exceeds a threshold for $L$ consecutive frames:

$$\text{LOST}_t = \mathbb{I}\Big[\tilde{E}_{t-L+1:t} > \tau\Big], \tag{27}$$

where $\tilde{E}_{t-L+1:t} > \tau$ denotes $\tilde{E}_{t-k} > \tau$ for all $k = 0, \ldots, L-1$. This persistence constraint prevents oscillations between *tracked* and *lost* states. We use $L = 3$ for all experiments.

**Threshold Calibration.** We set $\tau$ *once* using a held-out validation split of our synthetic verification dataset (constructed as in Appendix H.3). Specifically, we compute scores for compatible pairs $\mathcal{D}_{in}$ and incompatible/failure pairs $\mathcal{D}_{out}$, and choose $\tau$ to satisfy a target false-positive rate on $\mathcal{D}_{in}$ (equivalently, a desired specificity), while maximizing recall on $\mathcal{D}_{out}$. Concretely, we choose $\tau$ as the smallest threshold that satisfies

$$\Pr(E > \tau \mid \mathcal{D}_{in}) \le 0.05, \tag{28}$$

and under this constraint select $\tau$ that maximizes $\Pr(E > \tau \mid \mathcal{D}_{out})$ on the validation split. This calibration uses no test-time labels and $\tau$ is fixed for all experiments and stress-test settings.

**Action upon Detection.** When $\text{LOST}_t = 1$, TOM triggers re-localization (SOM) to obtain a fresh mask and pose hypothesis. We then reset the memory pool $\mathcal{M}_t$ using the newly re-initialized crop. Otherwise, we update $\mathcal{M}_t$

*Table 9.* **Temporal policy suppresses transient false re-localizations.** Compatible sequences are perturbed with brief logit noise; lower is better.

| Noise | Binary | Temporal EMA+$L$ |
|---|---|---|
| 2 frames, std=1.5 | 80.50% | 2.70% |
| 2 frames, std=2.0 | 81.70% | 2.40% |
| 3 frames, std=1.5 | 80.50% | 2.80% |
| 3 frames, std=2.0 | 83.10% | 2.20% |
| 3 frames, std=3.0 | 89.40% | 2.50% |

only with frames that remain below the threshold (high-confidence tracking). We maintain a bounded memory pool of size $K = 16$ and update it with a FIFO policy: whenever a new high-confidence crop is added and the pool exceeds $K$, the oldest entry is removed.

## H.2. Temporal Robustness under Transient Noise

TOM is trained as a binary compatibility verifier, but STORM does not use its output as a naive per-frame re-localization trigger. Instead, the verifier logit is interpreted as a continuous compatibility signal and converted into the temporal energy-like policy described in Appendix H.1. This distinction is important in tracking: motion blur, transient occlusion, or crop jitter can cause short logit drops even when the tracker is still on the correct object.

To diagnose this behavior, we simulate compatible tracking sequences with brief injected logit perturbations. These sequences should remain tracked, so the metric is whether the policy incorrectly triggers LOST anywhere in the sequence. We compare a per-frame binary trigger against the temporal policy used by STORM, with $E = -g_\theta$, EMA smoothing, $\alpha = 0.9$, and an $L = 3$ consecutive-frame rule. As shown in Table 9, a per-frame binary decision is brittle in a tracking loop because a single noisy frame can trigger re-localization. The temporal policy suppresses these transient spikes, reducing false re-localization from roughly 80–89% to roughly 2–3%.

## H.3. Tracking Dataset

We construct the TOM training dataset from BOP datasets. For each object, we extract ground-truth masks to build a memory pool of clean segmented object crops, where the object is cut out using the GT mask (i.e., background suppressed). As shown in Fig. 9, each training example is a pair consisting of (i) an observation crop from a frame (a standard RGB crop around the object region) and (ii) a memory crop queried from the segmented pool. Positive samples pair an observation crop with a segmented memory crop from the same object identity, representing correctly tracked states for the classifier. Negative samples are generated in two ways: (i) identity confusion, by pairing an

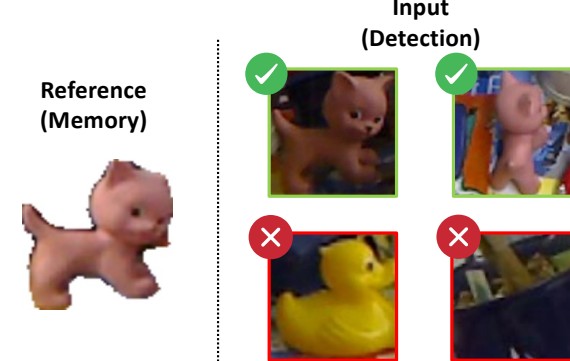

*Figure 9.* An overview of our tracking dataset. The task is to identify if the input from the detection model *i.e.* the tracked object matches the reference object, which is stored in the memory pool.

observation crop with a segmented memory crop from a different object; and (ii) tracking drift, by pairing the observation crop with an incorrect crop region, generated either by randomly shifting the crop window around the object region or by sampling random crop regions from the image. To avoid bias toward frequent objects, we sample object identities uniformly and generate 100,000 positive pairs, and then generate 100,000 negative pairs using the same sampling strategy, yielding 200,000 training pairs in total. This dataset design exposes TOM to both hard cross-object negatives and drift-like negatives, enabling it to learn subtle cues that distinguish successful tracking from failure cases.

## H.4. Stress Test on YCB-Video

We construct a stress-test benchmark on YCB-Video by applying two types of controlled perturbations to each sequence: (i) rapid motion simulated by frame dropping, and (ii) occlusion simulated by overlaying random masks on the object region. These perturbations intentionally break temporal continuity and induce tracking drift. Unless otherwise stated, all TOM hyperparameters (including the energy threshold $\tau$) are fixed once after validation and are not re-tuned for any corruption setting.

All results reported in the main paper use the *mild* stress-test setting, where both perturbations are applied with probability 0.1.

### H.4.1. RAPID MOTION VIA FRAME DROPPING

For each frame, we independently drop it with probability $p_{\mathrm{drop}} = 0.1$ (and keep it otherwise). The tracker is executed on the remaining frames in temporal order.

### H.4.2. OCCLUSION VIA RANDOM MASKS

For each frame, with probability $p_{\mathrm{occ}} = 0.1$, we overlay a single rectangular mask on the object region (defined

by the ground-truth object bounding box and mask). The rectangle area ratio $r$ is sampled uniformly relative to the object bounding-box area:

$$r \sim \mathcal{U}[0.10, 0.20], \tag{29}$$

and the aspect ratio is sampled uniformly from $[0.3, 3.3]$. The rectangle center is sampled uniformly from pixels inside the ground-truth object mask. Masked pixels are filled with zeros.

