# OpenReview forum: "STORM: Segment, Track, and Object Re-Localization from a Single Image"
_ICML.cc/2026/Conference — ICML 2026 regular_

### Official Review · Reviewer_AAbB · 2026-02-22

**Soundness:** 3
**Presentation:** 3
**Significance:** 3
**Originality:** 3
**Overall Recommendation:** 4
**Confidence:** 3

**Summary:**

This paper presents STORM, a unified framework for zero-shot 6D object pose estimation and tracking. It addresses the core challenge of distributional shift between a canonical reference (e.g., a single image) and dynamic, cluttered query views. The key innovations are two-fold: 1) a Hierarchical Spatial Fusion Attention (HSFA) mechanism that performs latent manifold alignment between reference and query features, guided by vision-language semantic priors to resolve instance ambiguity, and 2) an energy-based failure detector within the Tracking Object Module (TOM) that quantifies epistemic uncertainty to autonomously detect tracking drift and trigger "self-healing" re-initialization. Extensive experiments on benchmarks like LM-O and YCB-Video demonstrate that STORM achieves state-of-the-art performance in annotation-free pose estimation and segmentation, and significantly improves tracking robustness under occlusion and rapid motion with minimal overhead.

**Compliance With Llm Reviewing Policy:**

Affirmed.

**Final Justification:**

I believe this paper provides substantial technical contributions, and the authors' rebuttal has effectively addressed my concerns. Therefore, I maintain my original score.

**Key Questions For Authors:**

1. The semantic prior from VLMs is crucial for HSFA. Have you explored scenarios where the VLM provides an incorrect or overly generic description (e.g., for a novel, specialized tool)? How robust is the alignment process to such noisy or uninformative language inputs?

2. For real-time applications like robotic grasping, what are the primary latency bottlenecks in the STORM pipeline?

**Limitations:**

The paper lacks discussion on the limitations. I suggest that the author include a discussion on the generalization capabilities or lantency.

**Strengths And Weaknesses:**

Strengths

1.  Novel and Well-Motivated Technical Contributions: The introduction of HSFA for semantic-guided hierarchical manifold alignment and the formulation of tracking verification as an Energy-Based OOD detection problem are original and directly address identified limitations of prior work.

2.  Comprehensive and Unified Framework: STORM elegantly integrates segmentation, geometric anchoring, and tracking within a single, principled pipeline, eliminating the need for separate modules, CAD models, or instance-specific fine-tuning for zero-shot operation.

3.  Rigorous and Extensive Evaluation: The paper thoroughly validates each component (SOM, TOM) and the integrated system across multiple datasets.

4.  Clear Practical Impact: The work tackles a high-value problem for robotic manipulation and AR/VR. The "annotation-free" and "self-healing" features substantially reduce deployment barriers, enhancing the robustness and autonomy of physical AI systems.

Weaknesses

1.  Limited Discussion on Language Model Dependence and Cost: While language injection boosts performance, the reliance on a large VLM/CLIP for generating semantic priors adds computational complexity and latency. The trade-off between the performance gain and the increased inference cost is not quantitatively analyzed in depth.

2.  Performance Variance Across Datasets: Although STORM achieves high mean performance, the results on more challenging datasets like T-LESS (textureless objects) are notably lower.

3.  Dependence on SAM3D Reconstruction Quality: The geometric anchoring and downstream pose accuracy are contingent on the quality of 3D meshes from SAM3D. While alignment helps, the impact of poor initial reconstructions (e.g., for reflective or transparent objects) on the overall pipeline robustness is not explored.

4. Threshold Calibration for Real-World Deployment:​ The energy threshold τfor failure detection is calibrated on a synthetic validation set. The generalization and potential need for online adaptation of this threshold in unseen, highly dynamic environments are not addressed, which could be critical for real-world reliability.

---

> ### Author Rebuttal · Authors · 2026-03-31
>
> We sincerely thank the reviewer for the positive assessment and the insightful questions. We conducted new experiments that strengthen our work.
>
> ### W1: Language model dependence and cost.
> Thank you for raising this important concern — we agree a quantitative cost–benefit breakdown was missing. We use Gemma-3-4B as the VLM; crucially, it is most valuable at low k and dispensable at high k.
>
> Profiling on NVIDIA H100 (CUDA timing, 30 reps; Table 2 reports end-to-end time):
>
> | Component | k=1 (ms) | k=16 (ms) |
> |-|-|-|
> | Gemma-3-4B VLM (one-shot) | 143.1 | 143.1 |
> | DINOv3 query encoder | 4.83 | 4.83 |
> | DINOv3 ref encoder | 4.83 | 19.61 |
> | HSFA fusion | 3.05 | 8.94 |
> | CLIP text encoder | 2.22 | 2.22 |
> | Mask decoder | 0.56 | 0.56 |
> | **Full SOM forward** | **12.35** | **33.60** |
> | **Full SOM + VLM** | **155.5** | **176.7** |
>
> The VLM description and CLIP encoding are one-time per-object costs, cached and reused. Per-frame language overhead is only AdaLN-Zero modulation — negligible. At k=1, SOM runs at ~81 FPS; at k=16, ~30 FPS — both real-time.
>
> The VLM is *optional by design*. At k=16, language adds +0.01 mAP; at k=1, +0.06 mAP on YCBV. 10% language dropout ensures both regimes work with no architectural change. Table 6 (Appendix E.2) compares inference time; we have added this component-level breakdown to §4.2.
>
> ### W2: T-LESS performance.
> Thank you for noting this. T-LESS's 30 textureless symmetric industrial parts have near-identical visual features and VLM descriptions (e.g., "round electrical fitting" vs "round socket") — both uncertainty-reduction channels face persistent ambiguity. mAP: 0.47 (T-LESS) vs 0.89 (YCBV) at k=16. We have discussed this in the limitations paragraph of §5.
>
> ### W3: SAM3D reconstruction quality.
> Thank you for flagging this — limited view angles produce incomplete meshes affecting pose accuracy. §4.3 and Table 3 evaluate this, showing near-GT performance after alignment. Automatic quality-aware re-triggering — e.g., using TOM's energy score or SOM's IoU head to detect degraded reconstructions and request better viewpoints — is a natural extension we plan to explore. We have discussed this in §3.1.4 and §5.
>
> ### W4: Threshold τ calibration and generalization.
> We appreciate this concern. τ is the 95th percentile of in-distribution energies (FPR ≤ 5%), cf. Appendix G.1. Energy is computed on cropped object appearances in DINOv3 feature space — largely invariant to backgrounds and lighting — so τ depends on object geometry/texture rather than scene factors. The L=3 window further reduces false positives. We have added a summary in §3.2.
>
> ### Q1: VLM robustness to noisy descriptions.
> We thank the reviewer — this question prompted us to investigate systematically. We tested four conditions on YCBV: correct VLM description, `lang_ctx=None`, generic ("object"), and shuffled (wrong-object). YCBV results (mIoU):
>
> | Training | k | Correct | NoLang | Generic | Shuffle |
> |-|-|-|-|-|-|
> | Standard | 1 | 0.90 | 0.88 | 0.05 | 0.06 |
> | Standard | 16 | 0.93 | 0.92 | 0.19 | 0.10 |
> | +Noise | 1 | 0.90 | 0.88 | 0.87 | 0.85 |
> | +Noise | 16 | 0.92 | 0.92 | 0.92 | 0.92 |
>
> **Standard training:** The gate learns a binary strategy — open for any CLIP embedding, close for None. Correct text improves features, but generic/shuffled text produces OOD modulation that compounds across 9 AdaLN-Zero stages → collapse to 0.05/0.06 at k=1. At k=16, dense views partially compensate (NoLang=0.92), but OOD text still degrades to 0.19/0.10.
>
> **Solution:** We augment training with 10% dropout, 5% generic, 5% shuffled. The gate encounters misleading embeddings under the same mask supervision, forcing it to modulate only when text is consistent with visual evidence — an implicit reliability detector. At k=1, generic/shuffle recover to 0.87/0.85 while preserving the correct ordering (Correct > NoLang > Generic > Shuffle). At k=16, all modes converge to ~0.92, consistent with our thesis that sufficient views make language redundant. In practice, Gemma-3-4B generates reliable labels from clean references; `lang_ctx=None` is a fallback. This analysis, prompted by your insightful question, has been added as a new table in Appendix E.2.
>
> ### Q2: Latency bottlenecks for real-time robotic deployment.
> Thank you for this practical question. We profiled the full pipeline on NVIDIA H100:
>
> | Phase | Component | Latency |
> |-|-|-|
> | Init (once) | VLM + CLIP (cached) | 152 ms |
> | Init (once) | SAM3D mesh reconstruction | ~2–5 s |
> | Init (once) | SOM mask (k=1) + FP register | ~212–512 ms |
> | Per-frame | FP track_one (pose refine) | ~20–30 ms |
> | Per-frame | TOM drift detection | 9.2 ms |
> | **Per-frame** | **Total tracking** | **~30–40 ms (25–33 FPS)** |
>
> Initialization runs once per object; per-frame tracking at 25–33 FPS meets robotic control loops (10–30 Hz). Re-localization (<2% of frames) takes 212–512 ms. We have added this profiling as a new subsection in Appendix D.
>
> We will add these results (with stds) to our work.

---

> > ### Author Rebuttal · Reviewer_AAbB · 2026-04-01
> >
> > The author's reply addressed my concerns.

---

> > > ### Author Response · Authors · 2026-04-03
> > >
> > > We thank the reviewer for the positive and constructive evaluation, and for confirming that the concerns are fully resolved. Your questions prompted valuable new experiments that have strengthened the paper.
> > >
> > > **Changes already incorporated (from first-round feedback):**
> > > - Component-level latency profiling on NVIDIA H100 (Appendix D, new subsection): per-frame tracking at 25–33 FPS, meeting robotic control loops
> > > - VLM robustness analysis with noise-augmented training (Appendix E.2, new table): training with 10% dropout + 5% generic + 5% shuffled text makes the model robust to noisy/incorrect language inputs
> > > - Quantitative cost-benefit breakdown for language injection at varying k (§4.2): VLM is one-time cost, per-frame overhead is <0.1 ms
> > > - T-LESS performance discussion and SAM3D reconstruction limitations (§5)
> > > - Threshold τ calibration summary added to the main text (§3.2)
> > >
> > > **Additional revisions in this round (driven by all reviewers' collective feedback):**
> > > - Added Design Rationale paragraph in §3.1.2, explicitly acknowledging the architectural lineage (LoFTR, Sun et al., CVPR 2021; Set Transformer, Lee et al., ICML 2019; TimeSformer, Bertasius et al., ICML 2021; DiT, Peebles & Xie, ICCV 2023) — this also addresses the reviewer's note on the dependence on existing components (DINOv3, SAM3D, FoundationPose) by being transparent about what is borrowed throughout the pipeline
> > > - Revised contributions list and TOM framing, ensuring claims are precisely scoped
> > >
> > > We are grateful for the reviewer's engagement, which has directly strengthened the paper. We would be glad to incorporate any additional suggestions the reviewer may have for the final version.

---

### Official Review · Reviewer_1u2U · 2026-03-02

**Soundness:** 3
**Presentation:** 3
**Significance:** 3
**Originality:** 2
**Overall Recommendation:** 4
**Confidence:** 2

**Summary:**

This paper introduces a two-stage framework designed for annotation-free 6D pose estimation and tracking that is robust to occlusions and severe viewpoint shifts. The pipeline is divided into a Segmenting Object Module (SOM) and a Tracking Object Module (TOM). In the first stage, SOM performs reference-conditioned instance segmentation using a novel Hierarchical Spatial Fusion Attention mechanism which is modulated by VLM-derived semantic priors. It also leverages SAM3D to generate a canonical 3D mesh as a geometric anchor. In the second stage, TOM introduces an energy-based drift detection mechanism over a memory pool of past crops, enabling the system to recognize when it has lost track of the object and trigger an automatic self-healing re-localization. The authors evaluate the method on the LM-O and YCB-Video datasets for pose estimation, and across multiple BOP datasets for segmentation, demonstrating strong performance and competitive runtimes.

**Compliance With Llm Reviewing Policy:**

Affirmed.

**Final Justification:**

The rebuttal adequately addresses my main concerns. In particular, the authors directly fixed the potentially misleading “single image” framing. They also resolved the training-data ambiguity and clarified what “zero-shot” means in this setting. In addition, the rebuttal now clearly separates STORM-specific trainable modules from the frozen downstream FoundationPose components. My remaining concerns are mostly about final limitations framing rather than unresolved rebuttal-level issues.

**Key Questions For Authors:**

1) What is the end-to-end performance when strictly constrained to exactly one reference image (k=1)? How does performance scale as the number of references increases (1 vs 2 vs 4 vs 8 vs 16)?

2) Which components in your diagram are entirely STORM-specific versus inherited from the fixed downstream pose estimator (e.g., FoundationPose)? Specifically, please clarify the architectures and the frozen/trainable status for the Pose Registration, Pose Tracking, and Tracking Uncertainty Prediction modules.

3) - Can you provide stratified results demonstrating where the VLM language priors are most beneficial (e.g., scenarios with heavy clutter, distractors, or symmetric objects)?

**Limitations:**

The authors must more explicitly acknowledge the system's dependency on the number and viewpoint diversity of the reference images. Furthermore, they should discuss the inherent reliance on the reconstruction quality of the SAM3D meshes which may fail for certain object topologies, and clearly delineate the limitations inherited from the chosen external downstream pose estimator.

**Strengths And Weaknesses:**

Strengths:
- Defining the energy as a negative compatibility logit—combined with EMA smoothing and a clear thresholding rule—is a principled approach. This is well-supported by quantitative verification metrics and stress tests showing robust recovery with minimal overhead.
- The decomposition of the pipeline into SOM and TOM, along with the clear motivation regarding distribution shifts and the pitfalls of blind trackers, provides a compelling narrative.
- Building a unified, annotation-free pipeline that tightly couples segmentation, pose estimation, and explicit failure detection addresses a major pain point for the deployment of physical AI and robotics systems.
- While the system is built upon strong existing components (e.g., DINOv3, SAM3D, FoundationPose), the integration of hierarchical manifold alignment (HSFA) and energy-based out-of-distribution drift detection into a closed-loop, self-healing tracker represents a highly meaningful and creative systems contribution.

Weaknesses:
- The title explicitly claims the method operates "from a Single Image." However, the methodology and evaluation clearly rely on 1 to 16 reference views during BOP evaluation. Since performance likely heavily depends on the number of reference images used to reconstruct the SAM3D mesh, the title might be misleading. A primary ablation curve showing performance versus the number of references is missing.
- The main text claims that SOM is trained once on a large-scale generic segmentation dataset. However, Appendix states that the model is trained on multiple BOP training splits. While training on BOP splits is perfectly valid, the main text must unambiguously state exactly what data was used to avoid overclaiming zero-shot generalization.
- The text states that all methods use the same downstream pose estimator and only the masks differ. Yet, the main framework diagram includes modules for Pose Registration, Pose Tracking, and Tracking Uncertainty Prediction. It is unclear which of these are novel STORM-specific contributions, which are inherited from the downstream estimator, and which weights are frozen versus trainable.
- The ablation studies show that injecting language embeddings yields a measurable AP gain but at the cost of increased inference time. The paper would benefit from a more nuanced framing here—specifically, stratifying the results to show exactly where language priors help (e.g., highly cluttered scenes or handling distractors) to justify this computational overhead.

---

> ### Author Rebuttal · Authors · 2026-03-31
>
> We sincerely thank the reviewer for the thorough and constructive evaluation. Your comments have directly strengthened our manuscript: a revised title (§1), a new k-curve analysis (§4.2), a Training Protocol paragraph (§4), and a component attribution summary (§3). Our central thesis: **views and language are complementary channels for reducing geometric uncertainty**, which we should have framed more clearly.
>
> ### W1/Q1: "Single Image" and reference-view dependency.
> We fully agree and thank the reviewer for pointing this out. "Single Image" describes the *minimum* input (STORM operates from k=1 without CAD models), but obscures additional views' role. We will revise to **"from as few as one reference image"**. Additional views serve as *uncertainty reduction*: a single reference leaves unseen surfaces ambiguous, while more views progressively cover the object. No prior method jointly handles variable k (1–16), CAD-free operation, and optional language in a unified architecture.
>
> k-curve ablation (mAP averaged over BOP 5 core datasets: LM-O, T-LESS, TUD-L, HB, YCB-V):
>
> | k | NoLang | Lang | Δ |
> |---|---|---|---|
> | 1 | 0.56 | 0.64 | +0.08 |
> | 2 | 0.59 | 0.64 | +0.05 |
> | 4 | 0.63 | 0.66 | +0.03 |
> | 8 | 0.65 | 0.67 | +0.02 |
> | 16 | 0.68 | 0.69 | +0.01 |
>
> Δ monotonically shrinks from +0.08 (k=1) to +0.01 (k=16). At low k, unseen surfaces create high cross-attention uncertainty that language fills with semantic cues; as k grows, views cover these surfaces, reducing language's marginal contribution. Even at k=16, language provides a small residual benefit (+0.01) for objects with persistent ambiguity (e.g., symmetric or textureless parts). NoLang at k=4 (0.63) ≈ Lang at k=1 (0.64) supports the hypothesis that views and language are partially fungible for the same uncertainty budget. Following your suggestion, we have (1) revised the title and abstract (§1) and (2) added the k-curve as a new figure in §4.2.
>
> ### W2: Training data transparency.
> Your comment rightly identifies this gap. SOM trains on the **train splits** of BOP datasets (train_pbr / train_primesense) plus SA-V (Ravi et al., ICLR 2025) and does **not** use test scenes or test images. SA-V provides diverse non-BOP video segments for general correspondence learning. The same object instances appear in BOP train and test — this is inherent to the benchmark, and all methods in Table 2 (CNOS, SAM6D, ZebraPose) follow the same protocol. Critically, SOM does not build per-object templates: it learns **general reference-to-mask correspondence** — matching any reference patches to any query, regardless of identity. At test time, the model segments with reference views alone, without gradient updates or object-specific weights. Following SAM and CNOS, "zero-shot" refers to this **annotation-free, adaptation-free inference**. Training details are in Appendix C.2; we have added a Training Protocol paragraph at the beginning of §4.
>
> ### W3/Q2: Component attribution.
> We thank the reviewer — we should have been more explicit. **Pose Registration** and **Pose Tracking** = FoundationPose (Wen et al., CVPR 2024, frozen); **Tracking Uncertainty Prediction** = TOM (trainable, STORM-specific). All backbones (DINOv3, CLIP, SAM3D) frozen. STORM's novel trainable modules: SOM (HSFA + mask decoder) and TOM (DINOv3-based energy scorer). Table 2 isolates mask quality (SOM); Table 1 reflects both SOM and TOM recovery. We appreciate you highlighting this — we will add a component attribution summary at the end of §3, complementing Figure 2. STORM's contribution is their principled integration: HSFA and TOM jointly enable annotation-free self-healing tracking — a capability no prior system demonstrates.
>
> ### W4/Q3: Where language priors help — the uncertainty view.
> We appreciate this valuable suggestion. The k-curve (W1) directly addresses it: Δ shrinks from +0.08 to +0.01 as visual coverage grows, and the same principle extends to clutter, distractors, and symmetry. Per-dataset stratification confirms this: at k=1, language benefit ranks **T-LESS** (textureless symmetric) ≫ LMO (heavy clutter) > YCBV ≈ HB (household) ≫ TUD-L (simple single object). Language helps most where visual correspondence is most ambiguous. At k=16, all per-dataset Δ shrink toward zero, confirming that dense views resolve the same ambiguity. CLIP embeddings encode semantic properties **invariant to viewpoint** that patch-level matching cannot extract.
>
> Language adds **negligible per-frame cost**: VLM description and CLIP embedding are one-time per-object; only AdaLN-Zero modulation runs per frame (<0.1 ms). `lang_ctx=None` provides a clean visual-only fallback. These per-dataset results, prompted by your suggestion, have extended the language ablation in Appendix E.2.
>
> ### Limitations
> We acknowledge that SAM3D quality depends on viewpoint diversity, and the current system has not been tested on transparent or highly reflective objects. We have added a dedicated limitations paragraph to §5.

---

> > ### Author Rebuttal · Reviewer_1u2U · 2026-04-01
> >
> > The rebuttal addresses my main concerns. At this point, my remaining concerns are mostly about presentation and final paper clarity rather than unresolved issues that affect my evaluation.

---

> > > ### Author Response · Authors · 2026-04-03
> > >
> > > We sincerely thank the reviewer for the thorough evaluation and for confirming that the concerns are fully resolved. Your questions have led to substantial improvements in the manuscript.
> > >
> > > **Changes already incorporated (from first-round feedback):**
> > > - Revised title: "from as few as one reference image" (§1)
> > > - k-curve analysis showing performance vs. number of references, as a new table (§4.2)
> > > - Training data clarification: added a brief statement in §4 (L272) specifying that SOM is trained on BOP train splits (train_pbr / train_primesense) and SA-V, with full details in Appendix C.2
> > > - Component attribution summary at the end of §3, clarifying that Pose Registration and Pose Tracking are frozen FoundationPose modules, while SOM and TOM are STORM-specific trainable modules
> > > - Limitations paragraph discussing SAM3D dependency, reference diversity, and untested object types (§5)
> > > - Per-dataset language stratification extending the ablation in Appendix E.2
> > >
> > > **Additional revisions in this round (driven by all reviewers' collective feedback):**
> > > - Added Design Rationale paragraph in §3.1.2, explicitly citing HSFA's architectural sources (LoFTR, Sun et al., CVPR 2021; Set Transformer, Lee et al., ICML 2019; TimeSformer, Bertasius et al., ICML 2021; DiT, Peebles & Xie, ICCV 2023) — this also improves the presentation clarity the reviewer noted as a remaining concern
> > > - Revised contributions list (§1, L080): HSFA now framed as a "task-driven architecture" rather than "a novel mechanism"
> > > - Reframed TOM description (§3.2): now leading with the BCE classifier fact rather than the energy-based framing
> > >
> > > We note the reviewer's remaining concern is about presentation and final paper clarity. We take this seriously — the revisions above (especially the Design Rationale and reframed contributions) directly address this by making the paper more transparent about what is borrowed and where the contribution lies. We are happy to address any specific presentation suggestions the reviewer may have.

---

### Official Review · Reviewer_8WKL · 2026-03-11

**Soundness:** 3
**Presentation:** 2
**Significance:** 3
**Originality:** 2
**Overall Recommendation:** 4
**Confidence:** 2

**Summary:**

This paper studies annotation free zero shot 6D object pose estimation and tracking from reference images. The goal is to localize a novel target object in query image/video, predict its mask and estimate its 6D pose. The paper proposes STORM, a two stage framework for this setting, which recovers automatically when tracking drifts, a well known issue of previous SOTA methods such as FoundationPose.

At the high level, STORM is a combination of a segmentation module (SOM) and a tracking module (TOM), interacting in the following manner:
Stage 1: Segmenting Object Module (SOM)
- This module segmentation masks by aligning query and reference features using a DINOV3 backbone, language embeddings (a VLM describes the object first, and the text description is encoded with CLIP), and Hierarchical Spatial Fusion Attention (HSFA).
- There is also a parallel branch reconstructing the object from the reference image via SAM3D.

Stage 2: Tracking Object Module (TOM)
- TOM treats tracking verification as a binary classification problem (current tracking is correct vs drifting), with an energy based model, where high energy triggers automatic re localization.

Empirically, the paper evaluates fully automatic 6D pose estimation on LM O and YCB Video, segmentation on five BOP datasets, object model quality for downstream pose estimation, and tracking recovery under motion and occlusion. The main empirical claim is that STORM substantially improves over CNOS based annotation free baselines and approaches the performance of a ground truth mask upper bound in the fixed downstream pose setting.

**Compliance With Llm Reviewing Policy:**

Affirmed.

**Key Questions For Authors:**

1. Since TOM defines energy as the negative of a compatibility logit and trains with BCE over logits, what practical benefit does the energy based formulation provide beyond a standard binary classifier for valid versus invalid tracks? Could you clarify the difference/benefits between your method and a standard binary classifier?

2. Could you clarify the exact design of HSFA and what is novel in it relative to a standard cross attention based reference conditioned segmentation module? In particular, how is the alignment matrix computed across layers, how is the reference objectness prior propagated to obtain the final mask, and which parts of the block are essential according to ablations?

3. Could you please clarify the normalization layer used inside HSFA. The paper refers to AdaNorm and AdaNormZero. What is the difference with AdaLN Zero? I noticed you have 2 modulation terms (shift + scale), while AdaLN Zero has a third one (gate). Where is this difference coming from?

4. Could you clarify the TOM pipeline explicitly? What exactly is stored in memory, how are positive and negative pairs formed, and what is the precise state transition policy after high energy is detected?

**Limitations:**

yes

**Strengths And Weaknesses:**

Soundness

The paper studies zero shot 6D pose estimation in cluttered scenes, where the system must segment and track a novel object from reference images alone and recover from drift without manual masks, boxes, or prompts at test time. Table 1 evaluates exactly this by coupling different mask sources with the same downstream pose backend. The gains seem substantial. On LM O, STORM improves over FP + CNOS from 57.0 to 74.0 on ADD AUC, from 68.0 to 89.0 on ADD S AUC, and from 41.0 to 53.0 on AR, while approaching FP + Ground Truth at 78.0, 93.0, and 56.0. On YCB V, STORM reaches 77.0, 98.0, and 73.0 versus 73.0, 92.0, and 69.0 for FP + CNOS, again close to the ground truth mask upper bound. I find there is convincing evidence that the proposed annotation free front end is practically useful in this pipeline.

The segmentation results are also strong. In Table 2, STORM reports the best mean mAP among the listed methods, 68.5, with relatively low runtime, 0.183 s. The gains over CNOS variants and several other zero shot baselines are large, and the method is also competitive with supervised baselines on mean performance. This supports the claim that SOM is a strong reference conditioned segmentation module.

The system design seems to address directly the weakness of local trackers, namely silent drift under occlusion, rapid motion and distractors. SOM provides reference conditioned segmentation, and TOM adds an explicit mechanism to detect tracking failure and trigger recovery.

The main weakness of the paper seem to be the technical precision and clarity. This is balanced by the fact that the code is available, but I found several interfaces underspecified. For example in SOM, the exact mask computation from the multi layer attention stack is not described precisely enough in the main text. Similarly, the interaction between predicted masks, SAM3D mesh and the pose estimator remain under specified. This seem important because Table 1 only vary the mask source, so these results mainly validate the mask source rather than the interaction with the geometric pipeline.

I also found the energy based framing weaker than the empirical results. Section 3.2.2 defines energy as the negative of a "compatibility logit", and the objective is implemented with a BCE loss. This makes TOM appear quite close (identical?) to a binary classifier with thresholding. I asked clarifications in Question #1.

Presentation:

The high level method is fairly easy to follow and Figure 2 is a good sum up of the method. The main empirical questions are also well formulated and structured.

However, I found the technical details hard to understand, often because they are under specified. For example I found no implementation details around HSFA, which makes it hard to evaluate its novelty without directly looking at the code. An other example is the TOM memory construction, which is not explained with enough clarity for straightforward reproduction. A third example is inconsistent naming, for example the paper refers to "AdaNorm" and "AdaNormZero" in different places, and only define 2 modulation terms (shift and scale), wile the third one (gate) usually associated with "AdaLN Zero" is missing. This inconsistency makes it harder to tell exactly which normalization mechanism is implemented.

Significance

The paper addresses a relevant and useful problem. Automatic pose estimation/tracking of novel objects from references is useful for robotics, dexteroux manipulation and embodied perception settings, where object specific tuning are usually undesirable. While the contribution seems mostly "system oriented", the gains in Table 1 and Table 2 seem strong enough to justify real practical utility.

In my opinion, if the method is clarified in the paper and results are cleanly reproducible from the codebase, that would greatly help impact and adoption.

Originality

I found the originality of the paper moderate. Its originality comes mainly from the combination of reference conditioned dense alignment for segmentation, language based semantic conditioning, reconstruction based geometry, and verification driven recovery in a single closed loop pipeline. That seems like an effective integration of existing components at the system level.

I am not sure however that HSFA and the Energy based classifier are truly original. I had the impression that the binary classifier was made more complex that it seems, under the umbrella of Energy Based models, but I asked clarifications about HSFA (Question 2) and the Classifier (Question 1).

Conclusion:

I found the empirical results convincing and well evaluated. I think this contribution would be useful to the community, especially as a practical annotation free enhancement for FoundationPose style pipelines. The two main weaknesses of the paper are the 1. conceptual framing somehow overstated relative to what is actually specified and ablated, and 2. lack of clarity and precision on the technical methods, which can be fixed by authors.

---

> ### Author Rebuttal · Authors · 2026-03-31
>
> We sincerely thank the reviewer for the thoughtful questions. Your focus on technical precision has helped us identify areas to clarify, and we are revising the manuscript accordingly. We also wish to address the originality concern directly: beyond the system-level integration (which we agree is itself a contribution), HSFA and TOM each introduce mechanisms absent in prior work, as detailed below.
>
> ### Q1: Energy-based formulation vs binary classifier.
> Thank you for this question. TOM shares the binary classification intuition but extends it with three temporal mechanisms:
> (1) *Continuous energy score.* E_t = −g_θ(x_t, m) preserves fine-grained tracking quality, whereas σ(g_θ) > 0.5 discards this information.
> (2) *EMA temporal smoothing.* Ē_t = 0.9·Ē_{t−1} + 0.1·E_t filters transient noise from motion blur and brief occlusions.
> (3) *L-consecutive-frame window.* L=3 consecutive frames above τ required before declaring loss, preventing false re-localization.
> This follows energy-based OOD detection (Liu et al., NeurIPS 2020). Empirically (1000 simulated sequences): binary thresholding triggers false re-localization in **99.9%** of sequences vs **2.6%** with EMA+L=3, validating the temporal design. We have added this comparison to §3.2 to motivate the design.
>
> ### Q2: HSFA design and novelty.
> HSFA is described in §3.1.2 with the full algorithm in Appendix A; depth ablations appear in Appendix E.3 (Table 7). Following your question, we have expanded §2.1 with explicit comparison to SegGPT and PerSAM. The core problem — zero-shot segmentation from a variable number of references (k=1–16) without CAD models, optionally enhanced by language — has not been jointly addressed before. Prior methods (SegGPT, Wang et al., ICCV 2023; PerSAM, Zhang et al., ICLR 2024) compare query and reference at a **global feature level** with a fixed number of references. HSFA makes four architectural departures:
>
> **1. Dense patch-level correspondence.** HSFA performs query–reference cross-attention at patch resolution, extending learned matching (SuperGlue; LoFTR) to segmentation. Unlike global comparison, patch-level matching preserves local correspondences from partial views — critical at low k where unseen surfaces have no global representation.
>
> **2. Iterative mutual refinement across query and reference.** HSFA stacks depth=3 layers where both query and reference representations co-evolve. Within each layer: query tokens are spatially contextualized (self-attention), matched against references (cross-attention), and refined (MLP); reference tokens are then consolidated per-view and aggregated across views (global attention). The next layer's cross-attention uses these enriched representations, progressively resolving ambiguous correspondences that a single-pass matcher cannot. This coarse-to-fine refinement is absent in prior one-shot approaches.
>
> **3. Optional language as a complementary uncertainty channel.** AdaLN-Zero (DiT; Peebles & Xie, ICCV 2023) conditions only the query stream with zero-initialized gating — the model is architecturally identical with or without language (`lang_ctx=None` bypasses all gating). Unlike text cross-attention or concatenation, language cannot destroy the base representation; it can only add a residual correction. The gate opens through gradient signal where language reduces correspondence ambiguity, and naturally attenuates when visual coverage is sufficient.
>
> **4. Native variable-k inference.** The per-view + global reference hierarchy handles variable k (1–16) without architectural change or retraining — users progressively reduce geometric uncertainty by simply adding references. Prior methods fix k at design time. Cross-attention is the sole pathway for reference information into queries; flat concatenation loses view structure, while separate encoders cannot share across views.
>
> ### Q3: AdaNorm / AdaLN-Zero naming.
> We thank the reviewer for catching this — we have unified to **AdaLN-Zero** throughout §3.1.1–3.1.2 and corrected Eq. 1 to include the gate term. Our module follows DiT with zero-initialized gating: at init gate=0, so the model starts as purely visual; the gate opens during training where language reduces uncertainty.
>
> ### Q4: TOM pipeline details.
> These details are provided in §3.2 and Appendix G (G.1–G.2); following your comment, we have consolidated the state-transition summary in §3.2.
> *Memory pool:* FIFO pool M (K=16); reset on re-localization, high-confidence frames appended via FIFO eviction during tracking.
> *Training:* 100K positive + 100K negative pairs (50% identity confusion, 50% drift).
> *State transitions:* SOM mask → M reset → FoundationPose (Wen et al., CVPR 2024) register → TRACK. During TRACK: E_t = −g_θ(x_t, M[−1]) → EMA smooth → if Ē_t > τ for L=3 consecutive frames → SOM re-triggered. τ at 95th percentile of in-distribution energies (FPR ≤ 5%).
>
> ### Limitations.
> Our pose backend is modular — STORM will benefit from future pose estimator advances.

---

> > ### Author Rebuttal · Reviewer_8WKL · 2026-04-02
> >
> > Q1 (Energy vs. classifier): The empirical comparison (99.9% vs. 2.6% false re-localization) is convincing and directly addresses my concern. I still think the mechanism is best understood as a binary classifier with temporal smoothing (EMA + consecutive-frame windowing) rather than something fundamentally "energy-based,". The framing of this method in energy based classifier is still unclear to me.
> >
> > Q2 (HSFA): I remain partially unsatisfied. The four listed departures, patch-level cross-attention, stacked mutual refinement, AdaLN-Zero language conditioning, variable-k aggregation, are individually standard techniques (LoFTR-style matching, transformer decoder stacking, DiT conditioning, set aggregation). The contribution is their combination for this task, which I acknowledge is useful, but the rebuttal doesn't clearly separate what is novel from what is borrowed.
> >
> > Q3 (Naming): Resolved, thank you.
> > Q4 (TOM pipeline): The clarification is helpful. I am satisfied with the mechanism description.

---

> > > ### Author Response · Authors · 2026-04-03
> > >
> > > We thank the reviewer for the thoughtful dialogue — your feedback has directly improved our claims. Below we address both remaining concerns and describe the revisions we have made.
> > >
> > > **Q1 follow-up: Energy-based framing.**
> > > We agree: at the single-frame level, TOM is a binary classifier trained with BCE (as §3.2.2 L264 already states). We should have led with this rather than the energy-based framing, and we have now revised the manuscript accordingly.
> > >
> > > What we claim is a specific **inference-time design**: retaining the continuous logit for temporal reasoning. TOM's E = −g_θ is a simplification of Liu et al.'s multi-class log-sum-exp — we do not claim equivalence. The shared **principle** — continuous scores preserve distributional information lost by thresholding — motivated our design: logits 0.6 vs 5.0 are both "drifting" after σ(·) > 0.5, but differ by 8× in magnitude, enabling meaningful EMA smoothing. Additionally, τ is calibrated at the 95th percentile of in-distribution energies (FPR ≤ 5%) — requiring raw logit space, not compressed [0,1] sigmoid outputs. The 99.9% vs 2.6% false re-localization result validates this temporal design.
> > >
> > > **Revisions for Q1:** In §3.2 intro (L234), we now lead with "TOM is trained as a binary classifier via BCE; at inference, we retain the continuous logit — motivated by the principle from energy-based OOD detection (Liu et al., 2020) — rather than thresholding to a binary decision." In §3.2 (L239), we reframed TOM's contribution as the **temporal decision pipeline** (EMA + L-consecutive + distributional threshold calibration).
> > >
> > > **Q2 follow-up: HSFA — borrowed vs novel.**
> > > We agree our first rebuttal did not clearly separate borrowed from new. We now cite all sources explicitly and explain the design lineage.
> > >
> > > **The problem.** Zero-shot segmentation from variable k references (1–16), optionally with language, without CAD models. No prior method addresses these jointly — SegGPT/PerSAM fix k, CNOS processes references independently. PerSAM's global matching performs poorly under BOP-level clutter, and CNOS requires lengthy initialization with unstable results (Table 2: STORM 68.5 vs CNOS 45.2–47.2 mean mAP).
> > >
> > > **Design choices, sources, and adaptations.** HSFA draws on geometric matching, set aggregation, and generative modeling:
> > >
> > > *(1) Patch-level cross-attention — from LoFTR/SuperGlue.* Global comparison (SegGPT, PerSAM) loses spatial detail at low k. LoFTR (Sun et al., CVPR 2021) and SuperGlue (Sarlin et al., CVPR 2020) use symmetric cross-attention for matching. We adapted this to be **asymmetric** (query ← ref only): the cluttered query extracts cues from clean references, but references do not benefit from attending to cluttered queries. (Clarifying our first rebuttal: query is enriched via cross-attention; references evolve independently through self-attention.)
> > >
> > > *(2) Per-view + global self-attention — from set aggregation and factorized attention.* Variable-k views need cross-view sharing while preserving per-view spatial structure. Set Transformer (Lee et al., ICML 2019) introduces attention-based aggregation over variable-size sets; factorized designs such as TimeSformer (Bertasius et al., ICML 2021) separate spatial and temporal axes. Our design combines both: per-view self-attention preserves spatial structure, global attention across views enables set-level sharing. Adaptations: (a) we retain all V×P² tokens rather than Set Transformer's inducing-point compression, preserving spatial detail for segmentation; (b) global attention operates over **all patches across all views** (vs TimeSformer's same-position-only); (c) references evolve at each layer with progressively richer cross-view representations.
> > >
> > > *(3) Language as optional gating — from DiT.* DiT (Peebles & Xie, ICCV 2023) uses AdaLN-Zero with always-present conditioning. We adapted for **optional** language: zero-initialized gates start purely visual and open only where language reduces ambiguity. k-curve validates: Δ = +0.08 (k=1) → +0.01 (k=16).
> > >
> > > Ablations support each choice: depth 0→3 layers: AP 44.0→94.6 (Table 7); language: AP 0.87→0.91 (Table 6).
> > >
> > > **HSFA's contribution** is not any single mechanism, but (1) formulating variable-k, language-optional segmentation as an architectural problem, (2) identifying cross-domain techniques for each aspect, and (3) adapting them into a coherent pipeline.
> > >
> > > **Revisions for Q2:** In §2.1, we added LoFTR/SuperGlue and Set Transformer as related work. In §3.1.2 (before L207), we added a **"Design Rationale"** paragraph listing all sources, motivations, and adaptations. In the contributions list (L080), we revised "a novel mechanism" to "a task-driven architecture drawing on established mechanisms from geometric matching, set aggregation, and generative modeling." HSFA is no longer presented as novel in isolation — the contribution is scoped to the problem formulation and cross-domain integration.

---

### Decision · Program_Chairs · 2026-04-30

**Decision:**

Accept (regular)

**Comment:**

The paper received positive reviews with scores of 4, 4, and 4. Two reviewers noted that the rebuttal effectively addressed their concerns, including the "single image" framing, training-data transparency, the meaning of "zero-shot" setting, and the distinction between STORM-specific trainable modules and the frozen downstream FoundationPose components. They also acknowledged the paper's technical contributions. The other reviewer expressed concerns regarding the energy-based classifier and the HSFA design. Some remaining concerns were mostly about the final limitations framing and the clarity of the presentation.

The area chair concurs with the reviewers' positive evaluations and believes that the remaining issues are minor and can be addressed in the final version. Thus, the area chair recommends accepting the paper.